# LEARNING TO AGGREGATE: REINFORCEMENT LEARNING FOR GENERALIZED LABEL AGGREGATION

## ABSTRACT

The rise of large language models (LLMs) as annotators has introduced new opportunities and challenges for label aggregation in data annotation pipelines. While traditional aggregation methods are designed for human crowd workers with independent judgments, they fall short when applied to LLM-generated annotations that exhibit high correlation patterns and provide rich explanatory justifications. To address these challenges, we introduce RFAGG, a reinforcement learning framework that dynamically aggregates LLM annotations by jointly modeling both labels and their corresponding justifications. To train RFAGG, we construct the AGG dataset by collecting question-answer pairs generated by different LLMs across various datasets. Then, RFAGG first uses LLMs to generate multiple aggregation responses containing reasoning tokens and final answers for each input, and then uses our proposed aggregation reward functions to update the model via the policy optimization algorithm. Experiments demonstrate that RFAGG significantly outperforms classical and recent aggregation methods. Most notably, it serves as *a general aggregation model, generalizing well to out-of-domain and previously unseen tasks*. Despite being trained only on limited classification tasks, RFAGG achieves an average improvement of 2.45% on diverse objective tasks and 5.2% on the Alpaca 2.0 subjective task compared to its base model. We will publicly release the AGG dataset and our source code.

## 1 INTRODUCTION

Data annotation is a cornerstone of machine learning, providing the labeled datasets essential for training and evaluating models. Traditionally, this process has relied on human annotators, whose individual judgments are aggregated to produce reliable labels. Common aggregation techniques, such as majority voting and the Dawid-Skene model (Dawid & Skene, 1979), are designed under the assumption that annotators provide independent judgments and focus solely on the labels themselves. Recently, the advent of large language models (LLMs) as annotators has disrupted this paradigm, offering new opportunities but also presenting unique challenges.

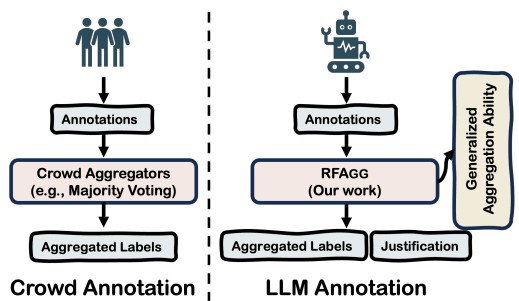

Figure 1: RFAGG enables effective aggregation of LLM annotations and achieves generalized aggregation ability beyond traditional methods.

LLMs, such as GPT (Achiam et al., 2023) and Qwen (Yang et al., 2024), have demonstrated remarkable capabilities in generating annotations across a wide range of tasks, from text classification to complex reasoning problems. Their scalability, cost-effectiveness, and ability to provide not only labels but also detailed justifications or explanations make them attractive alternatives to human annotators (Wang et al., 2024). However, aggregating annotations from LLMs is non-trivial. Unlike human annotators, LLMs often exhibit high correlation in their outputs due to shared training data and architectural similarities (Kim et al., 2025; Li, 2024b; Wang et al., 2024). Moreover, the rich justifications they provide are typically discarded in traditional aggregation methods, leading to a loss of valuable information that could otherwise enhance the aggregation process.

Using LLMs as aggregators is a promising approach, leveraging their advanced reasoning to synthesize information and generate coherent explanations Fan et al. (2024). However, raw aggregation ability does not correlate with general LLM performance (Chen et al., 2025; Li, 2024a). Without specialized training, these models are susceptible to prompt phrasing and majority-opinion bias, often failing to systematically resolve conflicts or fully utilize the explanatory content in justifications. Consequently, while employing LLMs as aggregators is appealing in theory, in practice, their raw aggregation performance often falls short of expectations and highlights the need for dedicated aggregation strategies.

A straightforward approach would be supervised fine-tuning (SFT) on reference aggregations. However, this paradigm is fundamentally limited. It trains a model to replicate a static ground-truth answer, teaching it *what* to generate but failing to provide a learning signal for *how* to reason through conflicting evidence and synthesize justifications. We therefore turn to reinforcement learning (RL). By directly rewarding desirable outcomes—such as accuracy and coherent conflict resolution—RL enables the model to learn a robust and generalizable aggregation policy, rather than simply mimicking a single reference text.

To this end, we propose RFAGG, a reinforcement learning framework that dynamically aggregates LLM annotations by jointly modeling both the predicted labels and their corresponding justifications. Unlike classical aggregation algorithms that focus solely on label consensus, RFAGG can achieve more effective and generalizable aggregation (Figure 1). Specifically, RFAGG formulates the aggregation task as a conditional text generation problem, where the goal is to generate an optimal aggregated annotation given an input instance and a set of LLM annotations. By leveraging reinforcement learning, RFAGG learns to generate aggregations that maximize a carefully designed aggregation reward function, which incorporates multiple facets of effective aggregation: accuracy, format consistency, and the ability to resolve complex cases with high annotator disagreement. To train and evaluate RFAGG, we construct the AGG dataset, a comprehensive collection of question-answer pairs from diverse datasets, annotated by multiple LLMs with varying personalization. The AGG dataset spans a wide range of tasks, including text classification, open-ended generation, arithmetic reasoning, and industrial visual inspection, ensuring that our evaluation captures the generalization capabilities of the aggregation methods.

Our contributions can be summarized as follows:

- We introduce RFAGG, a generalizable reinforcement learning-based framework for aggregating LLM annotations that jointly models labels and justifications, addressing the limitations of traditional aggregation methods.
- We construct the AGG dataset, which provides a rigorous benchmark for evaluating aggregation methods across diverse tasks and domains.
- We design the aggregation reward functions and a curriculum training strategy that together guide the model toward high-quality, consensus-driven aggregation by capturing the complexities of annotation justifications and progressively increasing task difficulty.
- Through extensive experiments, we demonstrate that RFAGG significantly outperforms both classical and state-of-the-art aggregation methods, achieving higher accuracy and robustness across various annotation tasks. Additionally, we show that RFAGG generalizes well to new tasks and domains, underscoring its potential as a universal aggregator.

## 2 RELATED WORK

**Label Aggregation.** Label aggregation is a critical step in handling data from multiple annotators (Zheng et al., 2017; Liu et al., 2023; Ustalov et al., 2024; Li et al., 2019b; Whitehill et al., 2009; Li et al., 2019a; Welinder et al., 2010; Zhou et al., 2012). Existing methods fall into three main categories: distance-based optimization methods, probabilistic graphical models (PGMs), and deep learning-based methods. Distance-based methods (Dawid & Skene, 1979; Aydin et al., 2014) focus on minimizing disagreement or divergence between annotator labels and estimated ground truth through iterative refinement. PGM-based methods (Li et al., 2019b;a) model the annotation process using probabilistic graphical structures that explicitly represent dependencies among workers, tasks, true labels, and latent variables. More recently, deep learning methods have emerged (Ratner et al., 2016; Liu et al., 2023), leveraging neural networks to capture complex, non-linear patterns in annotator

behavior and improve label inference accuracy. However, these methods are not designed for LLM annotators and typically ignore the rich justifications that LLMs can provide. In contrast, our method jointly models both labels and justifications, enabling richer aggregation and better utilization of LLM-generated signals.

**LLM as Annotators.** Recent research highlights LLMs' potential for data annotation tasks (Yao et al., 2024). GPT-3 demonstrated competitive performance with minimal fine-tuning in various NLP tasks (Gao et al., 2020), while Brown et al. showed LLMs could reduce the need for large task-specific datasets (Brown, 2020). He et al. introduced AnnoLLM, improving annotation performance through reasoned explanations (He et al., 2024a), and found GPT-4 could outperform traditional crowdsourcing methods (He et al., 2024b). Additional studies evaluated LLMs across specific annotation contexts, including political affiliation classification (Törnberg, 2023), implicit hate speech detection (Huang et al., 2023), general NLP tasks (Ding et al., 2022), paraphrase generation (Cegin et al., 2023), and replicating crowdsourcing pipelines (Wu et al., 2023). These findings collectively demonstrate LLMs' capabilities and limitations as data annotators across diverse NLP applications. Our framework differs by explicitly leveraging both label and justification information and by using reinforcement learning to optimize aggregation quality.

**Reinforcement Learning in Large Language Models.** In recent years, reinforcement learning (RL) has been central to advancing the alignment and performance of large language models (LLMs) (Cao et al., 2024), from alignment via RLHF and DPO (Christiano et al., 2017; Rafailov et al., 2023; Cao et al., 2024) to enhancing reasoning through verifiable rewards in large reasoning models (Plaat et al., 2024; Jaech et al., 2024; Guo et al., 2025). Methods like GRPO show that structured accuracy feedback can elicit planning and self-correction even before fine-tuning (Shao et al., 2024). Reinforcement learning naturally fits decision-making tasks by optimizing policies through rewards, enabling effective handling of conflicting evidence and coherent outcome generation. In contrast, supervised fine-tuning merely imitates labels without guiding reasoning. For aggregation, where models must synthesize inconsistent annotations, RL provides a clear advantage by directly rewarding accuracy and conflict resolution, yielding more robust and transferable policies.

## 3 METHODOLOGY

In this section, we present our RFAGG framework for dynamically aggregating LLM annotations using reinforcement learning. We first formalize the annotation aggregation task, then describe the construction of the AGG dataset, and finally detail our reinforcement learning method with specialized reward functions (Figure 2).

### 3.1 TASK FORMULATION

Let $x$ be an input instance (e.g., a text passage, an image, or a multimodal input) that requires annotation. We have $K$ LLMs (different LLMs or LLM with different personalization), denoted as $\{M_1, M_2, ..., M_K\}$, each providing an annotation $a_k = (l_k, j_k)$ for $x$, where $l_k$ is the label and $j_k$ is the justification or explanation for the label.

The goal of annotation aggregation is to combine these $K$ annotations into a single consensus annotation $a^* = (l^*, j^*)$ that maximizes accuracy and reliability. Traditionally, aggregation methods focus solely on the labels $\{l_1, l_2, ..., l_K\}$ while discarding the justifications. In contrast, RFAGG jointly models both labels and justifications to extract richer signals from the annotations.

We formulate this task as a conditional text generation problem, where given an input $x$ and a set of annotations $\{a_1, a_2, ..., a_K\}$, the goal is to generate an optimal aggregated annotation $a^*$. Formally:

$$a^* = \arg\max_a P(a|x, a_1, a_2, ..., a_K) \tag{1}$$

### 3.2 AGG DATASET

Although RFAGG was designed to aggregate annotations from multiple LLMs in order to produce consensus results that are both accurate and robust, we found that no existing publicly available dataset adequately supports this particular use case. Specifically, while existing benchmarks provide

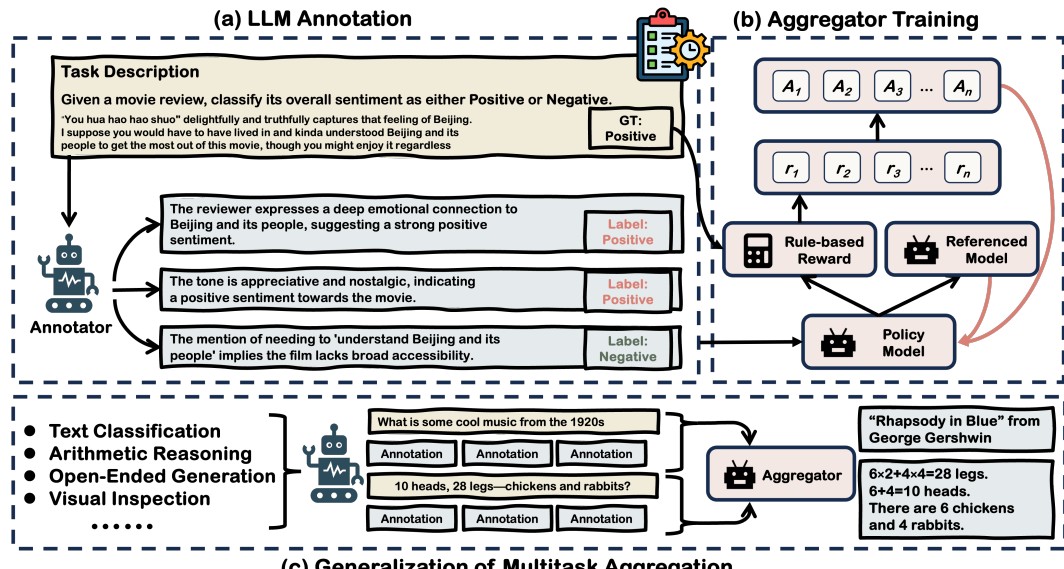

Figure 2: Framework of RFAGG , consisting of three main stages: (1) LLM-based annotation, (2) aggregator training, and (3) multitask aggregation generation.

input instances and ground-truth labels but lack the crucial component of multiple, diverse LLM-generated annotations. To address this gap and provide a comprehensive benchmark for evaluating RFAGG's performance, we constructed a dataset called AGG.

**Dataset Partition.** To rigorously assess RFAGG's aggregation capabilities and generalization, we designed a comprehensive dataset suite. This suite includes distinct partitions for training, testing, and generalization, facilitating a structured and thorough evaluation of both performance and robustness.

The training set and test set are primarily used to optimize and validate the aggregation ability of RFAGG. These sets were constructed from eight well-established benchmarks covering classification and multiple-choice tasks across diverse textual domains and difficulty levels. Specifically, these include: AGNews (Zhang et al., 2015b), DBPedia14 (Zhang et al., 2015a), Emotion (Saravia et al., 2018), IMDB (Maas et al., 2011), MultiNLI (Maas et al., 2011), SST2 (Socher et al., 2013), Trec (Li & Roth, 2002), WikiToxic (cjadams et al., 2017), and MMLU (Hendrycks et al., 2021). These datasets serve as prototypical examples of tasks where label aggregation methods are commonly employed. Moreover, they clearly defined categorical boundaries, which facilitates the derivation of precise reward signals during the reinforcement learning stages of RFAGG 's training The details of the datasets can be found in Appendix B.

The generalization set, on the other hand, is designed to evaluate how well RFAGG performs when applied to tasks and modalities outside the scope of its training distribution. To achieve this, we incorporated 5 heterogeneous datasets representing a broader spectrum of tasks: text classification (BoolQ (Clark et al., 2019) and WiC (Pilehvar & Camacho-Collados, 2019)), multiple-choice science questions (ARC (Clark et al., 2018)), arithmetic reasoning (GSM8K (Cobbe et al., 2021)), open-ended generation (Alpaca (Dubois et al., 2024)), and industrial visual inspection (MVTec AD (Bergmann et al., 2019)).

**LLM Annotation Generation.** All annotations in the AGG dataset were generated using two variants of the QWEN2.5 series: QWEN2.5-7B-INSTRUCT(Team, 2024a) for textual tasks and QWEN2.5-VL-7B-INSTRUCT (Team, 2025) for vision-language tasks. For each instance in every dataset, the LLMs annotator generated both a predicted label $l$ and an accompanying justification $j$ in natural language, providing reasoning for the selected label.

To generate a large and diverse set of annotations, we employed a highly scalable strategy by applying distinct personality prompts to a single base model. This method is significantly more practical for

creating extensive training datasets than relying on the finite and often limited selection of available LLMs. Meanwhile, we confirm the generalizability of our method in Appendix 4.6, which shows strong performance when aggregating outputs from different LLMs.

**Aggregation Dataset Construction.** To simulate real-world crowdsourcing scenarios, we constructed aggregation-ready datasets through subsampling of the generated annotations. For each input instance, a random subset of annotations was selected from the available variants. This method both preserves the stochastic nature of crowd annotations through varying annotation set sizes and mitigates potential biases related to set size in comparisons of aggregation methods.

### 3.3 RFAGG TRAINING FRAMEWORK

**Aggregation Reward Function.** The heart of the RFAGG framework is our meticulously designed, multi-faceted reward function. It is engineered not just to pursue correctness, but to sculpt the aggregator's behavior towards robust and intelligent synthesis (The MDP formulation can be found in Appendix C).

*Format Reward $R_{format}$.* Evaluates the format of the aggregation by checking whether the reasoning process and final answer are properly separated. Outputs that lack structure or fail to follow the expected format are penalized. This reward is computed using a rule-based parser, and the format reward is defined as:

$$R_{\text{format}}(O) = \begin{cases} 1, & \text{if } O \text{ has valid structure with labeled reasoning and answer} \\ 0, & \text{otherwise} \end{cases} \quad (2)$$

where $O$ denotes the aggregation model's output. This encouraging the model to maintain a consistent and interpretable structure in its aggregations.

*Accuracy Reward $R_{acc}$.* Measures the accuracy of the final aggregated answer by extracting the answer portion from the output using predefined rules, then comparing it to the gold standard label via exact match. This reward is also based on rules: let $y_{pred}$ be the extracted answer and $y*$ be the gold label. Then,

$$R_{\text{acc}} = \mathbb{I}(y_{pred}, y*) \quad (3)$$

where $\mathbb{I}(\cdot)$ is the indicator function. This reward provides the primary signal for guiding the model toward factually correct aggregations.

*Annotation Complexity Reward $R_{complex}$.* Assigns additional reward for successfully handling high-entropy cases where annotator disagreement is substantial. The entropy of annotations $H(A)$ is calculated as:

$$H(A) = -\sum_{l \in L} p(l) \log p(l) \quad (4)$$

where $L$ is the set of unique labels and $p(l)$ is the proportion of annotators who assigned label $l$. Higher entropy indicates greater disagreement, and solving such complex cases receives an additional reward scaled by the entropy:

$$R_{complex} = R_{acc} \cdot \max(0, H(A) - \tau) \quad (5)$$

where $\tau$ is a threshold parameter that determines what level of entropy qualifies as a complex case.

This reward formulation explicitly encourages the model to: (1) maintain coherent structure in its outputs, (2) maximize predictive accuracy, (3) excel at resolving difficult cases with high annotator disagreement.

**Reinforcement Learning with Curriculum-based training.** For policy optimization, we select Group Relative Policy Optimization (GRPO) (Shao et al., 2024) due to its demonstrated stability and sample efficiency in complex text generation tasks. Its value-function-free nature reduces computational overhead and simplifies the training pipeline, allowing us to focus our efforts on the more critical aspects of our framework: the reward engineering and curriculum design. We initialize our aggregation model using a pre-trained LLM (e.g., Qwen-2.5-7B) and proceed directly to reinforcement learning without supervised fine-tuning. The GRPO objective function is given by:

$$J_{\text{GRPO}}(\theta) = \mathbb{E}_{q \sim P(Q), \{o_i\}_{i=1}^{G} \sim \pi_{\theta_{\text{old}}}(O|q)} \left[ \frac{1}{G} \sum_{i=1}^{G} \frac{1}{|o_i|} \sum_{t=1}^{|o_i|} \left\{ \min \left( \frac{\pi_\theta(o_{i,t}|q, o_{i,<t})}{\pi_{\theta_{\text{old}}}(o_{i,t}|q, o_{i,<t})} \hat{A}_{i,t}, \right. \right.$$
$$\left. \left. \text{clip} \left( \frac{\pi_\theta(o_{i,t}|q, o_{i,<t})}{\pi_{\theta_{\text{old}}}(o_{i,t}|q, o_{i,<t})}, 1 - \epsilon, 1 + \epsilon \right) \hat{A}_{i,t} \right) \right] - \beta \mathbb{D}_{KL}[\pi_\theta || \pi_{old}] \right\}, \tag{6}$$

where $q$ is the query (input and annotations), $\{o_i\}_{i=1}^{G}$ are the generated candidate aggregations, $\pi_\theta$ is the current policy, $\pi_{\theta_{\text{old}}}$ is the old policy, $\hat{A}_{i,t}$ is the advantage, $\epsilon$ is a hyperparameter that controls the clipping and $\beta$ controls the strength of the KL term.

The advantage $\hat{A}_{i,t}$ is calculated based on the relative performance within a group of samples:

$$\hat{A}_{i,t} = \frac{r_i - \mu_r}{\sigma_r}, \tag{7}$$

where $r_i$ is the reward for candidate $i$, $\mu_r$ is the mean reward across all candidates in the group, and $\sigma_r$ is the standard deviation of rewards within the group. This method eliminates the need for a separate value function, reducing computational overhead.

For curriculum learning, we pre-calculate the difficulty of each instance based on the number of annotations to be aggregated:

$$\text{difficulty}(x) = |A_x| \tag{8}$$

where $|A_x|$ is the number of annotations available for input instance $x$. This metric serves as a direct measure of aggregation complexity, as instances with more annotations require the model to reconcile a greater number of potentially conflicting viewpoints.

We partition the training dataset into subsets of increasing difficulty before training begins:

$$\mathcal{D}_k = \{(x, \{a_1, a_2, \dots, a_{|A_x|}\}, y) \in \mathcal{D} \mid |A_x| = k\} \tag{9}$$

where $\mathcal{D}_k$ is the subset of data with exactly $k$ annotations per instance. The curriculum progresses from datasets with fewer annotations to those with more annotations:

$$\mathcal{D}_{\text{curr}} = \mathcal{D}_{k_{\min}} \cup \mathcal{D}_{k_{\min}+1} \cup \dots \cup \mathcal{D}_{k_{\text{curr}}} \tag{10}$$

where $k_{\min}$ is the minimum number of annotations and $k_{\text{curr}}$ is the current maximum number of annotations included in training.

This method ensures that the model first masters simpler aggregation problems with fewer annotations before progressing to more complex cases where it must reconcile a larger number of potentially conflicting annotations. The pre-calculation of difficulty levels based on annotation count allows for a structured curriculum without the need for dynamic sampling during training.

## 4 EXPERIMENT

### 4.1 EXPERIMENT SETUP

We conduct all experiments using the QWEN2.5-7B-INSTRUCT model as our base model. Model training is performed on the training split of the AGG dataset, while evaluation is conducted on both the held-out test split and a dedicated generalization split to assess out-of-domain performance. This protocol enables a rigorous examination of the model's aggregation ability as well as its robustness and transferability to novel tasks and domains.

To comprehensively evaluate RFAGG, we benchmark against three categories of baselines: (1) *Raw annotation*, where predictions from individual LLM annotators are used without any aggregation;

Table 1: Accuracy results on the AGG test set. The top section displays annotation performance generated by QWEN2.5-7B-INSTRUCT. The middle section presents the aggregation results from traditional label aggregation methods. The bottom section reports the performance of aggregation using LLMs. F1 score can be found in Appendix E

| Dataset | AGNews | DBPedia14 | Emotion | IMDB | MultiNLI | SST2 | TREC | WikiToxic | MMLU |
|---|---|---|---|---|---|---|---|---|---|
| ANNOTATION | 80.0 | 94.8 | 58.0 | 95.0 | 82.8 | 93.1 | 81.4 | 78.8 | 72.0 |
| MV | 80.7 | 96.5 | 59.4 | 95.5 | 85.1 | 93.8 | 84.8 | 79.8 | 75.0 |
| PM | 80.3 | 96.4 | 58.8 | 95.0 | 85.5 | 93.6 | 83.6 | 81.6 | 72.0 |
| DS | 81.7 | 96.8 | 59.0 | 95.3 | 85.3 | 93.9 | 87.4 | 78.5 | 76.0 |
| CATD | 80.7 | 96.4 | 59.2 | 95.2 | 85.8 | 93.6 | 83.4 | 81.1 | 75.0 |
| BWA | 80.6 | 96.8 | 59.3 | 95.1 | 85.6 | 93.6 | 85.6 | 81.2 | 76.0 |
| IBCC | 81.6 | 96.8 | 59.1 | 95.3 | 85.5 | 93.9 | 86.8 | 78.5 | 76.0 |
| EBCC | 80.1 | 96.8 | 58.3 | 95.2 | 85.1 | 93.9 | 87.0 | 79.0 | 76.0 |
| QWEN2.5-7B | 81.3 | 96.7 | 57.7 | 95.4 | 85.5 | 94.1 | 85.4 | 80.9 | 72.0 |
| RFAGG | **89.2** | **98.3** | **65.0** | **96.0** | **86.8** | **96.1** | **91.8** | **85.7** | **78.0** |

(2) *Traditional label aggregation methods*, including Majority Voting (MV), Probabilistic Model (PM) (Aydin et al., 2014), Dawid-Skene (DS) (Dawid & Skene, 1979), Confidence-Aware Truth Discovery (CATD) (Li et al., 2014), Bayesian Weighted Averaging (BWA) (Li et al., 2019a), Independent Bayesian Classifier Combination (IBCC), and Enhanced Bayesian Classifier Combination (EBCC) (Li et al., 2019b), which are applied to classification datasets; and (3) *Pretrained LLM aggregation*, where the QWEN2.5-7B-INSTRUCT model is used as an aggregator without any additional tuning.

## 4.2 AGG TEST SET PERFORMANCE ANALYSIS

We first evaluate the effectiveness of aggregation by comparing all methods with raw LLM annotations across the AGG test set (Table 1). Across most datasets, we observe that aggregation methods consistently yield substantial performance gains over using unaggregated annotations, confirming the value of synthesizing multiple LLM-generated responses. Among all evaluated methods, our proposed RFAGG achieves the best performance on every dataset, surpassing all baselines.

Among the baselines, we observe that simple majority voting (MV) provides a strong starting point, consistently improving over individual annotations. However, more sophisticated statistical aggregation methods such as PM, DS, and EBCC do not reliably outperform MV. This finding suggests traditional aggregation methods are less effective with LLM annotators due to their highly correlated outputs. Unlike human-generated data, LLM annotations don't exhibit the independent reliability variations that these aggregation techniques were designed to exploit.

We further assess the use of untuned LLMs as aggregators (QWEN2.5-7B-INSTRUCT), and observe that their aggregation performance generally falls short of MV, underscoring the limitations of relying on general-purpose LLM reasoning for this specialized task. In contrast, our proposed RFAGG achieves the best performance across all datasets, surpassing both traditional and LLM-based baselines. These results demonstrate that our reinforcement learning framework, which explicitly models both label and justification information, is highly effective at extracting consensus from LLM-generated annotations and robustly improves aggregation quality.

## 4.3 GENERALIZATION PERFORMANCE ANALYSIS

In this section, we evaluate RFAGG on the AGG generalization set to rigorously assess its out-of-domain generalization capabilities. Crucially, none of the tasks in this evaluation set are present in the training data, and several tasks feature fundamentally different characteristics (e.g., open-ended questions instead of classification tasks).

**Generalization on text classification.** Table 2 presents results on the classification datasets BoolQ and WiC. Across both benchmarks, RFAGG outperforms all other aggregation methods on both datasets, achieving the highest accuracy across these benchmarks. These results demonstrate the strong generalization ability of our framework to classification tasks not seen during training.

Table 2: Generalization accuracy comparison on classification dataset.

| Dataset | BoolQ | WiC |
|---|---|---|
| ANNOTATION | 82.2 | 66.4 |
| MV | 83.3 | 69.2 |
| PM | 83.6 | 65.6 |
| DS | 84.0 | 69.2 |
| CATD | 83.1 | 66.8 |
| BWA | 83.3 | 68.1 |
| IBCC | 84.0 | 69.4 |
| EBCC | 83.0 | 69.1 |
| QWEN2.5-7B | 83.3 | 68.6 |
| RFAGG | **86.8** | **70.6** |

Table 3: Generalization accuracy comparison on reasoning and math datasets.

| Dataset | ARC | GSM8K |
|---|---|---|
| ANNOTATION | 88.7 | 72.8 |
| MV | 90.8 | 86.0 |
| PM | 90.8 | - |
| DS | 90.8 | - |
| CATD | 90.8 | - |
| BWA | 90.8 | - |
| IBCC | 90.8 | - |
| EBCC | 90.8 | - |
| QWEN2.5-7B | 88.0 | 87.0 |
| RFAGG | **91.8** | **88.0** |

Table 4: Generalization performance result on AlpacaEval 2. The top section shows results from state-of-the-art LLMs, obtained from the official leaderboard. The middle presents annotations from QWEN2.5-7B-INSTRUCT and the bottom section compares LLMs aggregation.

| Metrics | LC Win Rate(%) | Win Rate(%) |
|---|---|---|
| CLAUDE 3 OPUS (02/29) | 40.5 | 29.1 |
| LLAMA 3.1 405B INSTRUCT | 39.3 | 39.1 |
| GPT-4 | 38.1 | 23.6 |
| QWEN2 72B INSTRUCT | 38.1 | 29.9 |
| LLAMA-3.1-70B-INSTRUCT-TURBO | 38.1 | 29.9 |
| QWEN1.5 72B CHAT | 36.6 | 26.5 |
| CLAUDE 3 SONNET (02/29) | 34.9 | 25.6 |
| LLAMA-3-70B-INSTRUCT | 34.4 | 33.2 |
| MISTRAL LARGE (24/02) | 32.7 | 21.4 |
| MIXTRAL 8X22B V0.1 | 30.9 | 22.2 |
| ANNOTATION | 28.1 | 31.0 |
| QWEN2.5-7B | 35.9 | 52.9 |
| RFAGG | **41.1** | **55.3** |

**Generalization on reasoning and math.** Table 3 shows results on a multiple-choice QA reasoning dataset (ARC) and a math problem-solving dataset (GSM8K). RFAGG consistently achieves the best performance across both benchmarks, underscoring its robustness and stability across tasks of varying domains and complexity. Notably, traditional methods like PM and DS are unsuitable for GSM8K due to their reliance on categorical outputs, underscoring the flexibility of RFAGG in handling diverse output formats within a unified aggregation framework. On the ARC multiple-choice task, many traditional aggregation methods (PM, DS, CATD, etc.) do not improve upon a simple majority vote. This is because, with a limited number of annotators exhibiting similar performance, these models struggle to estimate meaningful differences in reliability, often causing their predictions to converge with the majority opinion.

**Generalization on open-ended tasks.** We further assess RFAGG on open-ended tasks using AlpacaEval 2, with results summarized in Table 4. The results first highlight the inherent value of aggregation for open-ended tasks; the untuned QWEN2.5-7B model, when used as a simple aggregator, already improves performance over the raw annotations. Crucially, RFAGG outperforms this base aggregator, achieving a 5.2% gain in LC Win Rate and a 2.4% improvement in Win Rate. Compared to annotations generated by state-of-the-art LLMs, our method surpasses state-of-the-art LLMs including CLAUDE 3 OPUS, GPT-4, and LLAMA 3 405B INSTRUCT in both metrics, demonstrating strong generalization in open-ended scenarios, along with significantly enhanced aggregation capability.

Table 5: Generalization accuracy result on DS-MVTec.

| Subtask | Anomaly | Defect | | | | Object | |
|---|---|---|---|---|---|---|---|
| | Detection | Analysis | Classification | Description | Localization | Analysis | Classification |
| ANNOTATION | 77.0 | 81.1 | 63.3 | 66.2 | 65.2 | 85.8 | 92.7 |
| MV | 80.4 | 87.9 | 69.4 | 73.4 | 72.0 | 91.1 | **98.0** |
| PM | 79.8 | 83.4 | 67.9 | 68.0 | 67.7 | 87.8 | 97.1 |
| DS | 70.6 | 89.5 | 71.1 | 73.4 | 72.0 | 91.3 | **98.0** |
| CATD | 80.6 | 88.5 | **70.2** | 73.4 | **72.2** | 91.1 | 97.8 |
| BWA | 80.4 | 88.2 | **70.2** | 73.4 | 71.8 | 91.2 | **98.0** |
| IBCC | 81.1 | 89.6 | **70.2** | 73.4 | **72.2** | 91.3 | **98.0** |
| EBCC | 81.4 | 88.1 | 69.4 | 72.8 | 71.5 | 91.3 | **98.0** |
| QWEN2.5-7B | **82.5** | 90.9 | 70.1 | 73.8 | 70.3 | 91.3 | 93.5 |
| RFAGG | 78.0 | **91.9** | 68.6 | **74.6** | **72.2** | **91.8** | 96.1 |

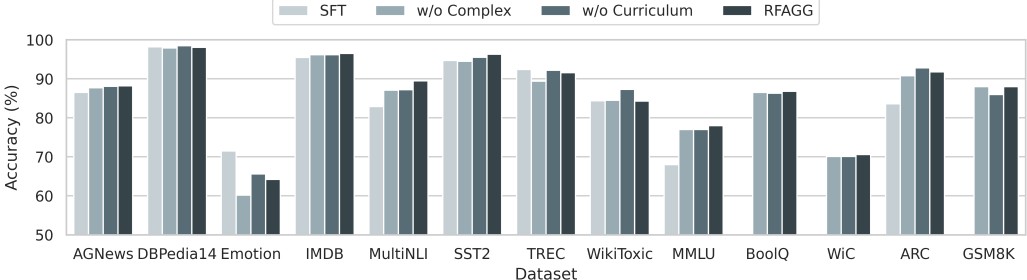

Figure 3: Ablation study results on classification, multiple-choice, and arithmetic reasoning datasets from both test and generalization splits.

**Generalization on industrial visual inspection.** Finally, Table 5 shows that RFAGG achieves the highest accuracy on four out of seven DS-MVTec tasks. While traditional methods like CATD and IBCC perform competitively in this domain, RFAGG delivers superior overall performance, demonstrating strong generalization in real-world industrial applications.

Taken together, these results demonstrate the remarkable generalization ability of RFAGG. For a wide range of tasks, including text classification, complex reasoning, open-ended generation, and visual inspection, RFAGG consistently outperforms both traditional statistical aggregation methods and untuned LLM-based aggregators. This strong out-of-domain performance confirms that RFAGG can serve as a unified and adaptable aggregation solution for diverse LLM annotation scenarios, accommodating various data modalities, task formats, and levels of annotation heterogeneity.

## 4.4 ABLATION STUDY

To better understand the contributions of each component in our framework, we conduct a series of ablation studies by evaluating three distinct variants of RFAGG. The first variant, *SFT*, replaces our reinforcement learning objective with supervised fine-tuning directly on ground-truth labels, aligning the model through standard supervised signals. The second variant, *w/o Complex*, removes the entropy-based component from the reward function $R_{complex}$, eliminating the mechanism that prioritizes high-disagreement instances. The third variant, *w/o Curriculum*, disables the curriculum learning strategy, training the model without progressively increasing task complexity.

The results, summarized in Figure 3, highlight the importance of each component. While the *SFT* variant achieves strong performance on the test set, it fails to generalize and yields zero accuracy on datasets requiring outputs that differ from those seen during training, providing strong empirical validation for our core premise: the task of generating reasoned aggregations is fundamentally ill-suited for SFT and necessitates the RL paradigm we propose. Both the *w/o Complex* and *w/o Curriculum* variants remain competitive on certain benchmarks but exhibit diminished overall and out-of-domain performance compared to the full model. These findings demonstrate that the entropy-

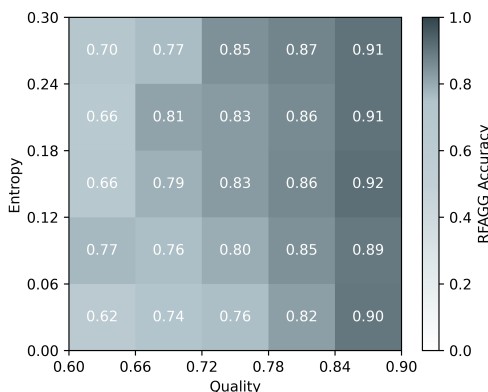

Figure 4: Aggregation performance related to annotation quality and diversity.

Table 6: Aggregation performance using annotations from diverse LLMs

| Method | Accuracy |
|---|---|
| Annotation | 65.0 |
| MV | 74.0 |
| DS | 73.0 |
| PM | 73.0 |
| CATD | 64.0 |
| BWA | 74.0 |
| IBCC | 74.0 |
| EBCC | 72.0 |
| Qwen2.5-7B-Instruct | 75.0 |
| RFAGG | **77.0** |

based reward and curriculum learning are crucial for improving both aggregation quality and the generalization ability of RFAGG, and that each component plays a complementary role in the robustness of our method.

### 4.5 DIVERSITY AND QUALITY ANALYSIS

Figure 4 analyzes how aggregation performance is influenced by both the quality and diversity of annotation outputs, evaluated across all annotators combinations. The x-axis denotes average answer accuracy (quality), and the y-axis indicates normalized label entropy (diversity). Each cell shows the mean aggregation accuracy. Our analysis shows that higher annotation quality and increased diversity both contribute to better aggregation performance. This finding aligns with the conclusion of (Kuncheva & Whitaker, 2003) that ensembles with diversity above a certain threshold offer guaranteed improvements over individual model. In the context of LLM annotation aggregation, this suggests that when LLMs generate a heterogeneous range of responses, our framework is better equipped to reconcile disagreements and synthesize a more accurate and robust consensus label.

### 4.6 GENERALIZATION ON DIFFERENT LLMS

To evaluate the robustness and generalization of our aggregation method, we construct a diverse and challenging testbed using annotations from ten distinct LLMs: including Daredevil-8B (Dar), Falcon3-7B-Instruct (Team, 2024b), Hermes-2-Theta-Llama-3-8B (Teknium et al.), Ministral-8B-Instruct-2410 (Min), OLMo-2-1124-7B-Instruct (OLMo et al., 2024), Qwen2.5-7B-Instruct (Team, 2024a), Yi-1.5-9B-Chat (Young et al., 2024), c4ai-command-r7b-12-2024 (Cohere et al., 2025), deepseek-llm-7b-chat (DeepSeek-AI, 2024), and openchat-3.6-8b-20240522 (Wang et al., 2023). All models generate annotation answers on the MMLU dataset.

As shown in Table 6, RFAGG achieves the highest accuracy, outperforming both classical aggregation rules and direct LLM-based aggregation. This demonstrates that RFAGG effectively leverages heterogeneous annotations and generalizes well across diverse model behaviors.

## 5 CONCLUSION

We introduced RFAGG , a reinforcement learning framework for aggregating LLM annotations by modeling both labels and justifications. To enable robust evaluation, we constructed the AGG dataset featuring diverse, quality-controlled question-answer pairs from multiple LLMs across various tasks. Experiments show that RFAGG consistently outperforms classical and recent aggregation methods, and generalizes well to new domains, highlighting its promise as a universal annotation aggregator.

## ETHICS STATEMENT

The datasets used in this work are derived from publicly available sources and constructed using open large language models. No private or sensitive data is involved.

## REPRODUCIBILITY STATEMENT

To ensure full reproducibility, we will release both the datasets and trained models upon publication.

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

APPENDIX

## A    THE USE OF LARGE LANGUAGE MODELS (LLMs)

During the writing process, we utilized LLMs, specifically GPT-5, to refine the manuscript's language for clarity and fluency. The authors retained full responsibility for all content, with the LLMs serving exclusively as a tool for language enhancement.

## B    AGG DATASET

Table 1 details the data distribution across splits. For the MMLU, AI2ARC, and GSM8K datasets, test sets are constructed following the sampling protocol introduced by Polo et al. (2024). For the DS-MVTEC visual inspection pairs, we adopt the question–ground truth annotation framework introduced in Jiang et al. (2025).

Table 1: Number of samples in the training, test and generalization set.

| Dataset | | | #Train | #Test |
|---|---|---|---|---|
| AGNews | | | 2,000 | 1,000 |
| DBPedia14 | | | 2,000 | 1,000 |
| Emotion | | | 2,000 | 1,000 |
| IMDB | | | 2,000 | 1,000 |
| MultiNLI | | | 2,000 | 1,000 |
| SST2 | | | 2,000 | 872 |
| TREC | | | 2,000 | 500 |
| WikiToxic | | | 2,000 | 1,000 |
| MMLU | | | 10,000 | 100 |
| BoolQ | | | - | 1,000 |
| WiC | | | - | 1,000 |
| AI2ARC | | | - | 100 |
| GSM8K | | | - | 100 |
| AlpacaEval 2 | | | - | 805 |
| DS-MVTec | Anomaly | Detection | - | 1,691 |
| | Defect | Analysis | - | 1,205 |
| | | Classification | - | 1,205 |
| | | Description | - | 1,212 |
| | | Localization | - | 1,193 |
| | Object | Analysis | - | 1,367 |
| | | Classification | - | 464 |

## C    MDP FORMULATION

The aggregation task is formulated as a single-episode MDP for each input instance.

- State ($s_t$): The state at timestep t consists of the input instance x, the full set of K annotations $a_k$, and the sequence of tokens generated so far for the aggregated answer, $y_{<t}$.

- Action ($a_t$): The action is to select the next token $y_t$ from the model's vocabulary.

- Transition: The transition is deterministic: generating token $y_t$ moves the agent to the next state where the generated sequence is $y_{\leq t}$.

- Reward (R): The reward is sparse and is calculated only at the end of the generation episode (when an end-of-sequence token is produced). The total reward R is the sum of the components defined in Section 3.3.1: R = $R_{format} + R_{a}cc + R_{c}omplex$. This formulation treats the entire generation of an aggregated response as a single trajectory that receives a terminal reward.

# D  MACRO F1 SCORE FOR CLASSIFICATION

Table 2: Macro F1 Score on classification datasets

| Dataset | AGNews | DBPedia14 | Emotion | IMDB | MultiNLI | SST2 | TREC | WikiToxic | MMLU | BoolQ | WiC |
|---|---|---|---|---|---|---|---|---|---|---|---|
| ANNOTATION | 79.9 | 94.7 | 51.4 | 94.9 | 82.5 | 93.1 | 80.9 | 67.3 | 71.9 | 81.9 | 65.7 |
| MV | 80.6 | 96.4 | 52.7 | 95.4 | 84.8 | 93.8 | 86.2 | 68.0 | 74.6 | 83.0 | 68.5 |
| PM | 83.3 | 96.2 | 51.3 | 94.9 | 85.3 | 93.6 | 84.1 | 69.7 | 71.4 | 83.3 | 63.5 |
| DS | 81.7 | 96.7 | 51.9 | 95.2 | 85.1 | 93.9 | 88.7 | 67.0 | 75.8 | 83.7 | 68.7 |
| CATD | 80.6 | 96.2 | 52.2 | 95.1 | 85.5 | 93.6 | 83.9 | 69.2 | 74.8 | 82.8 | 65.9 |
| BWA | 80.5 | 96.7 | 52.3 | 95.2 | 85.3 | 93.6 | 87.0 | 69.5 | 75.8 | 83.0 | 67.1 |
| IBCC | 81.6 | 96.7 | 52.1 | 95.2 | 85.3 | 93.9 | 88.1 | 67.0 | 75.8 | 83.7 | 68.9 |
| EBCC | 79.9 | 96.7 | 51.8 | 95.1 | 84.9 | 93.9 | 88.3 | 67.5 | 75.8 | 82.7 | 68.3 |
| QWEN2.5-7B | 81.0 | 96.6 | 50.9 | 95.3 | 85.4 | 94.1 | 85.3 | 69.2 | 71.5 | 82.9 | 68.6 |
| RFAGG | **88.3** | **98.1** | **55.5** | **96.4** | **89.2** | **96.3** | **92.4** | **73.2** | **77.8** | **86.3** | **69.3** |

# E  ADDITIONAL RESULTS

**Comparison with Different LLMs for Aggregation.**   We evaluate the effectiveness of various large language models as aggregation modules, selecting models with different architectures and parameter scales: QWEN2.5-7B, QWEN2.5-14B (Team, 2024a), MISTRAL-7B (Jiang et al., 2023), and GEMMA-3-12B (gem). As shown in Table 3, while most LLM-based aggregation methods improve over direct annotation, model size or general reasoning capability does not directly correlate with aggregation performance. Notably, the 14B-parameter QWEN2.5 underperforms its 7B variant on multiple datasets, suggesting that larger models do not necessarily yield better consensus. Similarly, MISTRAL-7B and GEMMA-3-12B show inconsistent gains, highlighting the instability of off-the-shelf LLMs in aggregation tasks. In contrast, our method, RFAGG, achieves consistent and significant improvements across all datasets, demonstrating that effective aggregation requires more than raw model capacity. This result validates the necessity and novelty of our approach.

**Comparison with Post-LLM Strategies.**   We further compare our method with several state-of-the-art post-hoc multi-model reasoning techniques: SELF CONSISTENCY (Wang et al., 2022), DEBATE STYLE (Du et al., 2023), and LLM-AS-JUDGE (Dubois et al., 2024). These methods represent popular paradigms for refining model outputs through internal reasoning or inter-model interaction. As summarized in Table 4, while these strategies offer modest improvements, their performance varies significantly across datasets. In contrast, RFAGG outperforms all baselines on every dataset, achieving robust and consistent results. This demonstrates the superiority of our designed aggregation mechanism in fusing diverse predictions reliably and effectively, without relying on unstable dialogue dynamics or generic judgment prompts.

Table 3: Accuracy on classification datasets from multiple LLMs aggregation

| Dataset | AGNews | DBPedia14 | Emotion | IMDB | MultiNLI | SST2 | TREC | WikiToxic | MMLU | BoolQ | WiC |
|---|---|---|---|---|---|---|---|---|---|---|---|
| ANNOTATION | 80.0 | 94.8 | 58.0 | 95.0 | 82.8 | 93.1 | 81.4 | 78.8 | 72.0 | 82.2 | 66.4 |
| QWEN2.5-7B | 81.3 | 96.7 | 57.7 | 95.4 | 85.5 | 94.1 | 85.4 | 80.9 | 72.0 | 83.3 | 68.6 |
| QWEN2.5-14B | 77.4 | 87.6 | 55.6 | 93.5 | 84.1 | 90.9 | 82.4 | 81.4 | 78.0 | 82.7 | 69.1 |
| MISTRAL-7B | 68.1 | 78.4 | 46.8 | 76.6 | 60.6 | 73.9 | 64.8 | 59.3 | 32.0 | 66.2 | 44.9 |
| GEMMA-3-12B | 80.8 | 96.7 | 58.3 | 95.5 | 86.2 | 93.4 | 85.8 | 80.2 | 76.0 | 84.6 | 68.1 |
| RFAGG | **89.2** | **98.3** | **65.0** | **96.0** | **86.8** | **96.1** | **91.8** | **85.7** | **78.0** | **86.8** | **70.6** |

Table 4: Accuracy on classification datasets from multiple post-LLM methods

| Dataset | AGNews | DBPedia14 | Emotion | IMDB | MultiNLI | SST2 | TREC | WikiToxic | MMLU | BoolQ | WiC |
|---|---|---|---|---|---|---|---|---|---|---|---|
| SELF CONSISTENCY | 80.7 | 96.5 | 59.4 | 95.5 | 85.1 | 93.8 | 84.8 | 79.8 | 75.0 | 83.3 | 69.2 |
| DEBATE STYLE | 82.8 | 97.3 | 55.8 | 95.3 | 78.2 | 94.3 | 84.0 | 87.6 | 84.5 | 67.0 | 62.6 |
| LLM-AS-JUDGE | 81.0 | 96.1 | 55.5 | 95.2 | 80.5 | 94.4 | 81.4 | 82.7 | 74.0 | 84.0 | 60.0 |
| RFAGG | **89.2** | **98.3** | **65.0** | **96.0** | **86.8** | **96.1** | **91.8** | **85.7** | **78.0** | **86.8** | **70.6** |

# F CASE STUDY

Table 5: Case study from IMDB

| Query | Given a movie review, classify its overall sentiment as either Positive or Negative. |
|---|---|
| | The Mad Monster starts in Dr. Lorenzo Cameron's (George Zucco) laboratory as he perfects his discovery of how to turn a human being into a vicious wolf like monster by injecting animal blood into a human subject who happens to be his dim-witted servant Petro (Glenn Strange), apparently he plans to put the serum at the disposal of the war department who will use it to create an unstoppable army of these monsters, the ultimate soldier! However, first things first as Dr. Cameron has his sights set on some sweet revenge on the people who dismissed his experiments, forced him to resign & subjected him to public ridicule. Dr. Cameron puts his plan into action & uses his monstrous creation to murder Professor Blaine (Robert Strange), in an unfortunate turn of events Dr. Cameron is unable to control the beast & is spotted by a local farmer Jed Harper (Eddie Holden) who spreads the news like wildfire, in another unfortunate coincidence a reporter named Tom Gregory (Johnny Downs) gets wind of the story & starts to investigate, he starts to suspect Dr. Cameron & since Gregory is going out with his daughter Lenora (Ann Nagel) he has plenty of opportunity to sniff around...¡br /¿¡br /¿Directed by Sam Newfield this is really low budget stuff from the 40's, even worse it's dull unoriginal low budget stuff. The script by Fred Myton drags the extremely thin premise out to almost 77 minutes which is far too long, there is no variety in the story & it's basically the same thing over & over. The character's are dull clichés, the mad scientist who conducts pointless experiments that create a monster, the fragile pretty daughter, the reporter who plays the hero & by pure coincidence happens to both be investigating the mysterious deaths said mad scientist is responsible for & is romantically involved with his daughter, the dumb servant, stupid idiotic police & stereotypical shotgun wielding farmers who are always accused of being drunk. This was probably clichéd even back in 1942! The film plods along at a fairly slow pace & director Newfield never manages to maintain or generate much in the way of excitement or atmosphere which is a bad thing. Technically the film isn't great, obviously the budget was minuscule & the mad monster itself looks lame resembling an old homeless wino who hasn't shaved for a few weeks & has had a pair of plastic joke shop fangs placed in his mouth. The black and white cinematography is basic & static like most films from this period while the good Dr. Cameron's laboratory consists of a couch & a table with a few sorry pieces of scientific equipment on top. The acting is stiff & wooden with Petro looking like he's on dope throughout the entire film, Zucco as the mad scientist doesn't convince & is forgettable. The Mad Monster is a pretty lame horror film, there is very little here to entertain although at least I made it through to the end in a single sitting which when I think about it is a bit of an achievement in itself! |
| Annotation 1 | **Justification:** The review primarily focuses on the film's shortcomings, using terms like "dull," "unoriginal," "drag," "thin premise," "dull clichés," "plods along," "lame," "lame horror film," and "very little here to entertain." The language used is predominantly negative, criticizing the plot, characters, acting, and technical aspects of the film. 
 **Label:** Negative |
| Annotation 2 | **Justification:** The review uses phrases such as "low budget stuff", "dull unoriginal", "drag the extremely thin premise", "no variety in the story", "dull clichés", "plods along at a fairly slow pace", "lame horror film", and "very little here to entertain". These phrases indicate a negative sentiment towards the movie. 
 **Label:** Negative |
| Annotation 3 | **Justification:** The review criticizes the film for being unoriginal, dull, and low-budget, with poor acting and technical quality. The reviewer points out the clichéd plot and characters, slow pace, and lack of excitement or atmosphere. While they mention making it through the film, the overall tone is highly negative. 
 **Label:** Negative |
| Annotation 4 | **Justification:** This review is clearly negative, as it criticizes the films script, pacing, characters, and technical aspects. It mentions that the film is dull, unoriginal, and cliché-ridden. The reviewer also notes that the acting is stiff and the monster looks unconvincing. The tone is dismissive and disappointed throughout the review. 
 **Label:** Negative |
| ... | |
| Aggregation | **Justification:** The review is overwhelmingly critical of the film, highlighting issues with the plot, characters, pace, and overall quality. **The answers from the contributors have consistently pointed out negative aspects.** 
 **Label:** Negative |

Table 6: Case study from MMLU

| | |
|---|---|
| **Query** | Including the bottom how many sides are on a square-based pyramid?
A. three
B. four
C. five
D. six |
| **Annotation 1** | **Justification:** The question asks about the sides of a square-based pyramid, including the base. A pyramid with a square base has four triangular sides plus the square base itself, making a total of five sides. This can be visualized or sketched to ensure accuracy.
**Label:** C |
| **Annotation 2** | **Justification:** A square-based pyramid has a square base, which contributes 4 sides. The pyramid also has 4 triangular faces connecting the base to the apex. Thus, the total number of sides is the 4 sides of the base plus the 4 triangular sides, making 8 sides in total, but the question specifically asks for the sides including the base, which is 4.
**Label:** B |
| **Annotation 3** | **Justification:** A square-based pyramid has a square base and four triangular sides. Therefore, it has a total of 5 sides including the base.
**Label:** C |
| **Annotation 4** | **Justification:** A square-based pyramid has a square base, which contributes 4 sides. Additionally, it has 4 triangular sides meeting at a common vertex. Therefore, the total number of sides is the 4 base sides plus the 4 triangular sides, totaling 5 sides.
- Option A (three) is incorrect as it does not account for all sides.
- Option B (four) is incorrect since it only counts the base sides.
- Option D (six) is incorrect because a pyramid does not have 6 sides.
- Option C (five) is the correct total as it includes all 4 base sides and the 1 apex side created by the triangles.
**Label:** C |
| **Aggregation** | **Justification:** A square-based pyramid has a square base (4 sides) and 4 triangular sides. Therefore, it has 4 + 1 base = 5 sides in total.
**Label:** C |

Table 7: Case study 1 from AlpacaEval 2

| Query | Mike's mother had four kids. Three of them are named Luis, Drake, and Matilda. What is the name of the fourth kid? |
|---|---|
| **Annotation 1** | The name of the fourth kid is Mike. The riddle states that Mike's mother had four kids, and three of their names are given as Luis, Drake, and Matilda. Since the question asks for the name of the fourth kid, and we are told that one of the children is Mike, it follows that Mike is the fourth child. |
| **Annotation 2** | Based on the information provided, the names of three of the four kids are Luis, Drake, and Matilda. The question does not provide the name of the fourth child, so the name of the fourth kid is not specified in the given information. |
| **Annotation 3** | The name of the fourth kid is Mike.
Here is the logic breakdown:
1. The question states that Mike's mother had four kids.
2. It provides the names of three of the kids: Luis, Drake, and Matilda.
3. The question asks for the name of the fourth kid.
Since the question specifies that Mike's mother had four kids and mentions three of them by name, it is clear that the fourth kid referred to is Mike himself. |
| **Aggregation** | The name of the fourth kid is Mike.
Here is the reasoning:
1. The question states that Mike's mother had four kids.
2. It provides the names of three of the kids: Luis, Drake, and Matilda.
3. Since the question is asking for the name of the fourth kid, it logically follows that the name of the fourth kid is Mike, as Mike is explicitly mentioned to be one of the children.
Therefore, the fourth kid is named Mike. |

Table 8: Case study 2 from AlpacaEval 2

| Query | For answering the chat, think of a funny response you can use.
+ how are you? |
|---|---|
| **Annotation 1** | How am I? I'm doing a barrel roll in my imagination right now! How about you? Are you ready to dance the waddle while sipping your epoxy smoothie? |
| **Annotation 2** | I'm doing a barrel roll! Because even robots need to show off sometimes! |
| **Annotation 3** | Well, I'm not actually feeling a whole lot because I'm just a computer program, but thank you for asking! How about you, are you having a comedic adventure today? |
| **Annotation 4** | How am I? I'm wrestling with a wormhole of inner thoughts and trying to squeeze them all into a single response. How about you? Are you navigating the complexities of your day with the grace of a professional tightrope walker, or are you more of a safety harness kind of person? |
| **Aggregation** | How am I? I'm doing a barrel roll in my code and cracking jokes with myself—do you know why a computer never goes to the beach? Because it has too many bytes to sand! How about you, are you riding the storm of your day or just feeling like a calm wave? |

Table 9: Case study from DS-MVTec

| Query | There is a defect in the object. How many areas of defect are present on the fabric border of the zipper?"
A. One.
B. Four.
C. Two.
D. Three. |
|---|---|

| Annotation 1 | **Justification:** Upon sizes and Comparing the query image with the reference image, there doesn't appear to be a consistent pattern that would suggest a single defect across both pieces. The edge on the left side of the query image seems less uniform but not clearly specified as a defined defect.
**Label:** A |
|---|---|
| Annotation 2 | **Justification:** Upon comparing, I notice that one area where there appears to be a break or missing section along the top edge of the fabric near the slider indicates a defect. This is consistent across both images.
**Label:** C |
| Annotation 3 | **Justification:** The main aspects are to focus on the fabric border conditions, in order to determine if there is any defect present. There seems to be a defect at the bottom left corner of the fabrics on both sides of the zipper, indicating two distinct defects. For a closer look, one may also focus on continuity of features, like the zipper teeth as they may indicate better continuity when not defective.
**Label:** C |
| Aggregation | **Justification:** Looking carefully at the image, there are two clearly visible defects on the fabric border of the zipper. One defect is at the bottom left and another is at the top right, near the slider.
**Label:** C |

# G PROMPT

## G.1 AGGREGATION PROMPTS

---

**System Prompt**

As a domain-agnostic response synthesizer, critically evaluate multiple contributor submissions through three analytical lenses: (1) factual accuracy verification, (2) contextual relevance assessment, and (3) logical coherence validation. Integrate the most credible elements into a unified response maintaining original reasoning traces.

---

**User Prompt**

Following is the Task Description and the answers from different Contributors.
[Task Description]
⟨⟨Task Description and Question Input⟩⟩

[Contributor 1]
⟨⟨Annotation from Contributor 1⟩⟩
[Contributor 2]
⟨⟨Annotation from Contributor 2⟩⟩
...
[Contributor n]
⟨⟨Annotation from Contributor n⟩⟩

⟨⟨Format Instruction⟩⟩

---

## G.2 PROMPT VARIANTS

We developed 10 task description prompt variants for each dataset.

Table 10: Variants of prompt applied to the AGNews fataset

| |
| --- |
| Categorize the given article into one of the four categories: World, Sports, Business, or Sci/Tech. Justify your choice with a brief paragraph highlighting key details from the text that support your decision. |
| As a journalist, analyze the article and determine whether it belongs to World, Sports, Business, or Sci/Tech. Provide a thorough explanation with specific details from the text that support your categorization. |
| Examine the article's content and decide whether it best fits into World, Sports, Business, or Sci/Tech. Summarize your reasoning in a concise paragraph, focusing on the most relevant details. |
| Determine which category—World, Sports, Business, or Sci/Tech—best describes the article. Support your decision with specific examples from the text in a brief paragraph. |
| As a news editor, review the article and classify it under one of the four categories: World, Sports, Business, or Sci/Tech. Explain your reasoning concisely, citing relevant information from the article. |
| Assess the article's main themes and determine whether it belongs to World, Sports, Business, or Sci/Tech. Discuss how its content aligns with your chosen category and why it does not fit the others. |
| Analyze the article's primary focus, audience, and key elements such as stakeholder involvement, risk factors, or technological relevance. Then, classify it under World, Sports, Business, or Sci/Tech and justify your decision with specific details. |
| Evaluate all four categories—World, Sports, Business, and Sci/Tech—and explain why the article fits best into one of them. Provide a reasoned argument backed by key details from the text. |
| Identify possible categories for the article and discuss why each could be relevant or not. Based on this analysis, assign it to World, Sports, Business, or Sci/Tech and justify your final choice in a brief paragraph. |
| From a reader's perspective interested in global affairs, sports, financial markets, or technological advancements, determine whether the article belongs to World, Sports, Business, or Sci/Tech. Explain your choice using specific content references. |

Table 11: Variants of prompt applied to the DBPedia14 dataset

Using logical deduction, analyze the text and identify specific keywords or patterns that most strongly align with one of the following categories: Company, EducationalInstitution, Artist, Athlete, OfficeHolder, MeanOfTransportation, Building, NaturalPlace, Village, Animal, Plant, Album, Film, or WrittenWork. Then, select the category that best matches the context and nature of the text, explaining your reasoning step-by-step.

Apply a design thinking approach by first empathizing with the user who might be reading this text. Define what aspects of the text would resonate most with different categories such as Company, EducationalInstitution, Artist, Athlete, OfficeHolder, MeanOfTransportation, Building, NaturalPlace, Village, Animal, Plant, Album, Film, or WrittenWork. Ideate potential categories based on the tone, content, and purpose of the text. Prototype your solution by choosing the category that fits the text's core message. Test your hypothesis and explain the rationale behind your choice. Conclude with the selected category.

Utilizing first-principles reasoning, break down the text into its most basic elements. Analyze whether the core subject of the text aligns with the fundamental characteristics of the categories: Company, EducationalInstitution, Artist, Athlete, OfficeHolder, MeanOfTransportation, Building, NaturalPlace, Village, Animal, Plant, Album, Film, or WrittenWork. Starting from scratch, determine how these categories can be distilled into essential components and then reassemble them to select the most appropriate one. Output the final category clearly.

Engage in a comparative analysis of how different categories (Company, EducationalInstitution, Artist, Athlete, OfficeHolder, MeanOfTransportation, Building, NaturalPlace, Village, Animal, Plant, Album, Film, or WrittenWork) would interpret the text. For each category, outline how the text could be viewed differently by someone in that domain. Based on these comparisons, select the category that most appropriately reflects the essence of the text and provide justification for your decision. Output the chosen category.

Perform a deep dive into the text by taking a historical perspective. Consider how people from different time periods might have categorized the content. What historical context or shifts in understanding could influence the category selection today? Determine the category (Company, EducationalInstitution, Artist, Athlete, OfficeHolder, MeanOfTransportation, Building, NaturalPlace, Village, Animal, Plant, Album, Film, or WrittenWork) by considering this historical lens and output the final choice.

Frame the task as a hypothetical negotiation between experts in various domains. Each expert argues for their category (Company, EducationalInstitution, Artist, Athlete, OfficeHolder, MeanOfTransportation, Building, NaturalPlace, Village, Animal, Plant, Album, Film, or WrittenWork) based on the text's characteristics. You, as the mediator, must synthesize these arguments and select the most fitting category based on the strongest case made. Clearly output the selected category.

Apply a pattern recognition approach: search for specific linguistic patterns, terms, or concepts in the text that align with known markers of each category (Company, EducationalInstitution, Artist, Athlete, OfficeHolder, MeanOfTransportation, Building, NaturalPlace, Village, Animal, Plant, Album, Film, or WrittenWork). Cross-reference these patterns with typical representations found in each category to determine the most appropriate one. Conclude with the most fitting category.

Conduct a role-playing exercise where you take on the persona of an expert in one of the given categories (Company, EducationalInstitution, Artist, Athlete, OfficeHolder, MeanOfTransportation, Building, NaturalPlace, Village, Animal, Plant, Album, Film, or WrittenWork). Approach the text from that persona's perspective and argue why it should fall within that category. Afterward, switch roles to another category and repeat the process. Compare all arguments and output the final selected category.

Consider the task as if you are a curator of a museum that houses diverse exhibits, each representing one of the categories (Company, EducationalInstitution, Artist, Athlete, OfficeHolder, MeanOfTransportation, Building, NaturalPlace, Village, Animal, Plant, Album, Film, or WrittenWork). Based on the description and thematic elements in the text, decide which exhibit the text would most likely belong to, providing reasons for your curatorial decision. Output the category you selected.

Using a scientific method approach, formulate a hypothesis about the category (Company, EducationalInstitution, Artist, Athlete, OfficeHolder, MeanOfTransportation, Building, NaturalPlace, Village, Animal, Plant, Album, Film, or WrittenWork) based on an initial reading of the text. Collect evidence from the text to support or refute the hypothesis, and analyze it systematically. Based on this analysis, choose the category that best fits the content and context of the text. Output the final category.

Table 12: Variants of prompt applied to the Emotion dataset

| |
|---|
| Using logical deduction, analyze the given text and identify the underlying emotion by evaluating the tone, word choice, and context. Break down the components of the text systematically to determine which emotional category (Sadness, Joy, Love, Anger, Fear, or Surprise) best matches the overall expression. Classify it into one of these emotions. |
| Imagine you are a psychologist conducting a session with the person who wrote the text. Based on your understanding of human emotions, assess the emotional state they are likely experiencing, considering both explicit and implicit clues from the text. Classify the emotion as either Sadness, Joy, Love, Anger, Fear, or Surprise. |
| Perform a comparative analysis by taking the text and comparing it with known examples of emotional expression. Identify patterns in language and sentiment to match the text with one of the six emotional categories (Sadness, Joy, Love, Anger, Fear, or Surprise). Clearly classify the emotion based on the analysis. |
| Adopt a design thinking approach: empathize with the emotional context of the text, define the emotional challenge, ideate potential emotional states, prototype potential emotional interpretations, and test your hypothesis against the text. Conclude which emotion (Sadness, Joy, Love, Anger, Fear, or Surprise) fits best. |
| Utilizing first-principles reasoning, strip down the language and tone of the text to its fundamental emotional elements. From this minimalist analysis, deduce the emotion that fits the core essence of the text, clearly classifying it as Sadness, Joy, Love, Anger, Fear, or Surprise. |
| Frame this task as a negotiation between emotions. Imagine that the text is a statement from one party (the speaker), and each emotion (Sadness, Joy, Love, Anger, Fear, or Surprise) is a negotiator. Determine which emotion wins the negotiation and classify the text as that emotion. |
| Place yourself in the shoes of a poet interpreting the emotional subtext of the text. Using your creative intuition, deduce which emotion (Sadness, Joy, Love, Anger, Fear, or Surprise) is most aligned with the emotional landscape presented in the text and classify it accordingly. |
| Consider the emotional journey of a character in a story. The text you are analyzing is a snippet from their diary. Based on the tone and context, classify the emotion expressed as Sadness, Joy, Love, Anger, Fear, or Surprise. |
| Imagine that you are conducting a debate between a group of experts on emotional intelligence. The text is presented as evidence, and the experts are tasked with determining the emotion it represents. Based on the debate, clearly classify the emotion as Sadness, Joy, Love, Anger, Fear, or Surprise. |
| Imagine you are a detective analyzing the emotional clues in the text to solve a case. Investigate the language used, identify emotional triggers, and classify the emotion as either Sadness, Joy, Love, Anger, Fear, or Surprise based on the evidence found. |

Table 13: Variants of prompt applied to the IMDB dataset

Analyze the given movie review by identifying key phrases and emotional indicators. Focus on the tone, word choice, and expressions used to classify the sentiment as Positive or Negative. Clearly state the sentiment as either Positive or Negative.

Approach the review using a design thinking methodology. Start by empathizing with the reviewer's emotional journey, define the sentiment issue, brainstorm possible interpretations, and test the sentiment to classify it as Positive or Negative. Provide a final classification as Positive or Negative.

Apply first-principles reasoning to break down the sentiment of the review. Identify the fundamental emotional tones and descriptive language used, then synthesize them to determine if the overall sentiment is Positive or Negative. Conclude with a final sentiment classification as Positive or Negative.

Imagine a group discussion about the movie review. Simulate how different individuals might react to the review's language and tone to classify the sentiment as Positive or Negative based on their perspectives. Clearly state whether the sentiment is Positive or Negative.

Perform a comparative analysis of the review against reviews for similar films. Look for patterns in language, tone, and content to decide whether the sentiment is Positive or Negative, based on common conventions for the genre. Conclude with either Positive or Negative sentiment.

Assume the role of a mediator between two opposing sentiment analysts. One believes the review is Positive, the other Negative. Consider their arguments and come to a balanced conclusion about the review's sentiment. Explicitly state whether the sentiment is Positive or Negative.

Frame the movie review as part of a negotiation between the filmmaker and the audience. The filmmaker wants a Positive sentiment, and the audience's emotional response helps you classify the review as either Positive or Negative based on how it aligns with their expectations. Provide a final sentiment classification as Positive or Negative.

Incorporate a psychological approach by evaluating the emotional cues in the review. Consider the reviewer's emotional state and biases, then determine if the sentiment expressed is Positive or Negative based on those psychological factors. Clearly state the sentiment as Positive or Negative.

Adopt a linguistic analysis perspective. Focus on the nuances of word choice, sentence construction, and emotional undertones to classify the sentiment of the review as either Positive or Negative. Provide a conclusive sentiment classification.

Use a literary criticism approach by examining the review's rhetorical devices, narrative style, and overall tone. Identify whether the review's language leans more toward positivity or negativity to classify the sentiment as Positive or Negative. Explicitly provide the sentiment classification.

Table 14: Variants of prompt applied to the MultiNLI dataset

| |
|---|
| From a logical deduction perspective, carefully examine the premise and hypothesis. If the truth of the premise guarantees the truth of the hypothesis, label it as Entailment. If the premise neither supports nor contradicts the hypothesis, label it as Neutral. If the premise contradicts the hypothesis directly, label it as Contradiction. |
| Utilize a scenario-based approach: construct a scenario where both the premise and hypothesis could either align, be independent, or conflict. Based on the scenario's outcome, categorize the relationship as Entailment, Neutral, or Contradiction. |
| Apply a comparison strategy: break down the key elements of the premise and hypothesis. Determine if they logically reinforce one another (Entailment), are unrelated (Neutral), or present opposing ideas (Contradiction). |
| Use a first-principles approach: distill both the premise and hypothesis to their basic truths and foundational ideas. Based on these fundamentals, decide whether the relationship between them is one of Entailment, Neutral, or Contradiction. |
| Approach the task from a deductive reasoning perspective: systematically test whether the premise can be used to derive the hypothesis logically. If so, it's Entailment. If the premise doesn't influence the hypothesis, it's Neutral. If the premise disproves the hypothesis, it's Contradiction. |
| Frame the task as a debate. Treat the premise as one side of an argument and the hypothesis as the opposing argument. Determine if the arguments support each other (Entailment), exist independently (Neutral), or are in direct opposition (Contradiction). |
| Use a critical thinking methodology: identify and evaluate the core assumptions in the premise and hypothesis. Assess whether they complement (Entailment), do not affect each other (Neutral), or directly oppose one another (Contradiction). |
| Take a philosophical approach: engage with the premise and hypothesis through the lens of conceptual analysis. Does the premise inherently support the hypothesis (Entailment), allow for alternative interpretations (Neutral), or inherently refute the hypothesis (Contradiction)? |
| Adopt a systems-thinking approach. Consider the premise and hypothesis as parts of a larger system. Assess if they fit together naturally (Entailment), are disconnected (Neutral), or if one disrupts the integrity of the other (Contradiction). |
| Analyze the premise and hypothesis using a process of elimination. Test different logical possibilities where the premise could either lead to, be independent of, or contradict the hypothesis. Based on this elimination, classify the relationship as Entailment, Neutral, or Contradiction. |

Table 15: Variants of prompt applied to the SST2 dataset

| |
|---|
| Identify any specific words or phrases that indicate a positive or negative sentiment, and classify the sentence as 'Positive' or 'Negative' based on these indicators. |
| As a consultant tasked with identifying positive sentiment in movie reviews, break down the sentence into its core elements and analyze the language used for any indicators of satisfaction, enjoyment, or appreciation. Classify the sentence as 'Positive' or 'Negative' accordingly. |
| From the perspective of a strategist, apply first-principles reasoning to dissect the sentence and determine if it expresses positivity or negativity. Utilize fundamental principles of sentiment analysis to classify the sentence as 'Positive' or 'Negative'. |
| As an end-user reading a movie review, empathize with the reviewer and gauge the overall sentiment conveyed in the sentence. Based on your personal understanding, classify it as 'Positive' or 'Negative'. |
| Use foundational reasoning to identify basic indicators of sentiment and classify the sentence as 'Positive' or 'Negative' based on the logical structure of the words used. |
| Classify the sentence as 'Positive' or 'Negative' by carefully comparing it to both positive and negative reviews to identify the subtle differences that classify it. Consider the context and tone of the sentence in this comparison. |
| Classify the sentence as 'Positive' or 'Negative' by considering the nuances of sentiment, such as sarcasm and irony. Explicitly analyze the sentence for these indicators and justify your classification accordingly. |
| Simulate the context in which the sentence was written and the potential reactions it might evoke to classify the sentence as 'Positive' or 'Negative'. |
| Use a creative brainstorming approach to identify both overt and subtle indicators of sentiment, classifying them as either 'Positive' or 'Negative' and justifying your choices while emphasizing the importance of context and nuance. |
| Adopt a scenario-based thinking approach, imagining the scenarios described in the sentence to classify it as 'Positive' or 'Negative', and explain your reasoning. |

Table 16: Variants of prompt applied to the TREC dataset

| |
|---|
| Analyze the question carefully and determine which type of information is being requested: Is it describing something, asking for an entity, referring to a place, inquiring about a person, looking for a number, or using an abbreviation? Classify the question as Description, Entity, Location, Human, Numeric, or Abbreviation, and justify your choice. |
| Break down the question's structure. Does it focus on an object, a person, a place, a number, a description, or a shorthand term? After evaluating the elements, classify the question as Description, Entity, Location, Human, Numeric, or Abbreviation and explain your reasoning. |
| Think about the type of response the question is seeking: Does it ask for a label (Entity), a place (Location), a person (Human), a description (Description), a number (Numeric), or an abbreviation (Abbreviation)? Choose the correct category and provide a detailed justification for your decision. |
| Approach the question from a logical perspective: If the question seeks to identify an object, a location, a person, a numeric value, a descriptive fact, or an abbreviation, classify it accordingly. The correct category is: Description, Entity, Location, Human, Numeric, or Abbreviation. |
| Consider the question from a broader context: Does the inquiry fit into one of these categories: a factual description, an object or entity, a place, a person, a numerical value, or a shorthand form? Classify the question as Description, Entity, Location, Human, Numeric, or Abbreviation, and explain your rationale for the chosen category. |
| Imagine you're categorizing the question in a filing system designed for organizing knowledge. Which folder does it belong to—Description, Entity, Location, Human, Numeric, or Abbreviation? Choose the correct category and justify your reasoning. |
| Frame the question within a real-world scenario: Is it asking for information about an object (Entity), a place (Location), a person (Human), a number (Numeric), a description (Description), or shorthand (Abbreviation)? Classify the question as Description, Entity, Location, Human, Numeric, or Abbreviation and explain your reasoning. |
| Consider the purpose of the Query Is it inquiring about an object, a place, a person, a number, a description, or an abbreviation? Classify the question into one of these categories—Description, Entity, Location, Human, Numeric, or Abbreviation—and explain the reasoning behind your choice. |
| Take a systems-thinking approach: What category does the question best fit into based on the kind of information it's asking for? Classify the question as Description, Entity, Location, Human, Numeric, or Abbreviation, and offer a justification for your classification. |
| Imagine being tasked with explaining this question to someone unfamiliar with it. How would you categorize the question—based on whether it refers to a description, a specific entity, a place, a human, a numeric value, or an abbreviation? Provide the category: Description, Entity, Location, Human, Numeric, or Abbreviation, and explain your rationale. |

Table 17: Variants of prompt applied to the WikiToxic dataset

| |
|---|
| Using sentiment analysis, examine the given Wikipedia comment and categorize it based on the emotional tone conveyed. Label the comment as 'Toxic' if the sentiment is overwhelmingly negative, aggressive, or harmful, and 'Non-Toxic' if it is neutral or positive in tone. |
| Consider the linguistic structure and word choice within the comment. Identify specific toxic language indicators such as insults, threats, or discriminatory remarks. Classify the comment as 'Toxic' if it contains any of these features, and 'Non-Toxic' if it does not. |
| Adopt a scenario-based approach: imagine you are a moderator on a Wikipedia discussion page. Given the comment, would you consider it acceptable or offensive based on community guidelines? If it violates the guidelines, label it 'Toxic'; otherwise, label it 'Non-Toxic'. |
| Use a first-principles reasoning approach: Break down the comment to its fundamental components. Examine whether the language in the comment is objectively harmful or promotes toxicity, and classify it accordingly as 'Toxic' or 'Non-Toxic'. |
| Employ a comparative analysis methodology: analyze the given comment in the context of several other known toxic and non-toxic comments. Based on similarities in tone, phrasing, and content, decide whether the comment falls into the 'Toxic' or 'Non-Toxic' category. |
| Apply a design thinking approach: empathize with the potential audience of the comment. Analyze how the language might affect readers—whether it could harm, alienate, or provoke. If the language might cause harm, label it 'Toxic'; if it's neutral or beneficial, label it 'Non-Toxic'. |
| Frame this as a hypothetical negotiation: imagine you're negotiating the content standards for Wikipedia comments with a team of content moderators. Would this comment pass your collective standards? If it doesn't meet the agreed standards, classify it as 'Toxic'; if it does, classify it as 'Non-Toxic'. |
| Consider the comment from a psychological perspective: what motivations or emotions might be driving the language? If the comment reveals hostility, anger, or frustration aimed at others, classify it as 'Toxic'; otherwise, classify it as 'Non-Toxic'. |
| Think of the comment in terms of ethical considerations: if you were to act as a judge for the comment, would it be deemed as potentially harmful to the community? If the comment undermines respect and promotes harm, classify it as 'Toxic'; if not, classify it as 'Non-Toxic'. |
| Adopt a critical thinking approach: examine the logic and reasoning behind the comment. Is there evidence of personal attacks, stereotyping, or harmful assertions? If so, classify it as 'Toxic'; if it's logical, respectful, and constructive, label it as 'Non-Toxic'. |

Table 18: Variants of prompt applied to the MMLU dataset

| |
| --- |
| Assume the role of a logic professor tasked with solving this multiple-choice question using step-by-step logical deduction. Break down each option and explain the reasoning that leads to the correct choice. |
| Approach this multiple-choice question through first-principles reasoning. Strip the question down to its foundational concepts and build up your understanding to select the correct answer. |
| Use comparative analysis to evaluate all answer choices. Systematically compare and contrast the merits and flaws of each option to determine which is most accurate. |
| Adopt the mindset of a detective solving a mystery. Examine clues in the question and eliminate incorrect options using evidence-based reasoning to reveal the correct choice. |
| Frame the problem using systems thinking. Analyze how each component of the question interrelates and assess the systemic coherence of each option before identifying the correct answer. |
| Engage in reverse engineering. Assume each answer choice is correct, then work backward to see which one aligns logically with the conditions in the question. |
| Use design thinking by empathizing with the test-maker. Consider the intent behind the question and prototype different reasoning paths to arrive at the answer that best fits the underlying design. |
| Use a root cause analysis framework. Identify the core issue or concept tested by the question and trace it through each option to determine the most fitting response. |
| Analyze the question using analogical reasoning. Draw parallels between the current problem and known examples or models to infer the most applicable answer. |
| Assume the role of a debate moderator. Stage a brief internal debate between advocates for each answer choice and assess which argument most convincingly supports its claim. |

Table 19: Variants of prompt applied to the BoolQ dataset

| |
| --- |
| Apply logical deduction to analyze the given passage and identify if there is direct evidence supporting or denying the facts in the question. If the passage provides clear support for the facts, output 'Yes' If the passage contradicts the facts, output 'No' |
| Conduct a scenario-based simulation where you assume you are a fact-checker tasked with verifying the truthfulness of the statement in the question using the evidence provided in the passage. Based on the passage, output either 'Yes' if the facts are supported or 'No' if they are denied. |
| Use comparative analysis to contrast the facts presented in the question with the information found in the passage. If the passage aligns with the question, output 'Yes' If it contradicts or denies the facts, output 'No' |
| Think like a detective investigating the validity of a claim. Break down the passage and analyze whether there is sufficient evidence to support or refute the statement in the question. If supported, output 'Yes' If refuted, output 'No' |
| Adopt a design thinking approach and assess the passage for clarity, accuracy, and consistency with the question. If the passage confirms the facts in the question, output 'Yes' If it denies them, output 'No' |
| Approach the task through first-principles reasoning by deconstructing both the question and the passage to their most fundamental truths. If the passage logically supports the question's facts, output 'Yes' If the passage denies them, output 'No' |
| Imagine that you are a judge in a courtroom, and the question is being presented as evidence for a trial. If the facts in the passage confirm the question, output 'Yes' If the passage denies or contradicts the question, output 'No' |
| Conduct a thorough analysis of the passage as if you are an expert fact-checker. If the passage provides concrete evidence that supports the facts in the question, output 'Yes' If it denies or contradicts them, output 'No' |
| Frame the task as a negotiation between two parties: one party asserts the facts in the question, while the other denies them. If the passage supports the facts in the question, output 'Yes' If the passage denies the facts, output 'No' |
| Take on the perspective of a philosophical debate. Argue both sides of the claim in the question, and based on the passage, output 'Yes' if the passage supports the claim, or 'No' if the passage denies the claim. |

Table 20: Variants of prompt applied to the WiC dataset

| |
|---|
| Analyze the context in which 'w' is used in 's1' and 's2', and determine if both instances of 'w' convey the same core meaning, excluding any differences in form and tense. Output 'True' if they do, otherwise 'False'. |
| Use logical reasoning to determine if the core meanings of 'w' in 's1' and 's2' are consistent, excluding any differences in form and tense. Output 'True' or 'False'. |
| Determine if 'w' in both 's1' and 's2' has the same core meaning, excluding any differences in form and tense. Output 'True' or 'False'. |
| Consider the contextual understanding of 'w' in 's1' and 's2', and determine if it conveys the same core meaning, excluding any differences in form and tense. Output 'True' if they do, otherwise 'False'. |
| Analyze the meaning of 'w' in 's1' and 's2', and determine if they convey the same core meaning, excluding any differences in form and tense. Output 'True' or 'False'. |
| Examine the context in which 'w' is used in 's1' and 's2', and determine if both instances of 'w' convey the same core meaning, excluding any differences in form and tense. Output 'True' if they do, otherwise 'False'. |
| Act as a legal expert in a debate. Argue both sides of the case to determine if the word 'w' in 's1' and 's2' correspond to the same meaning. Output 'True' if the word's meanings are aligned, otherwise output 'False'. |
| Investigate how 'w' operates semantically in 's1' and 's2', ascertaining if the core meaning remains consistent when accounting for contextual usage. Output 'True' or 'False'. |
| Assume the role of a linguist and analyze the core meaning of 'w' in 's1' and 's2', excluding any differences in form and tense, and output 'True' if they do, otherwise 'False'. |
| In the role of a first-principles physicist, apply foundational principles to deduce the meaning of 'w' in 's1' and 's2'. If the principles align, output 'True'; otherwise, output 'False'. |

Table 21: Variants of prompt applied to the AI2ARC dataset

| |
|---|
| Solve the multiple-choice questions based on scientific reasoning and knowledge. |
| Tackle the set of science questions by choosing the appropriate multiple-choice answers. |
| Use your understanding of science to pick the correct answers from the given options. |
| Review and solve each science multiple-choice question logically. |
| Determine the right choice for every science question in the list. |
| Choose the most scientifically accurate response for every question presented. |
| Imagine you are a high school science teacher reviewing a quiz—select the correct answers for each student question. |
| Take the role of a science competition judge and evaluate each multiple-choice question for the best answer. |
| Act as a scientific researcher taking a certification exam—identify the right answers to maintain your credentials. |
| You're a tutor helping a student prepare for a science test. Choose the correct answers to explain to them later. |

Table 22: Variants of prompt applied to the GSM8K dataset

| |
|---|
| As a mathematics expert, carefully analyze the problem, justify each step rigorously, and ensure that your solution would be accepted in a formal mathematical journal. |
| As a student in a timed exam setting, solve the problem quickly and efficiently, balancing speed with accuracy, and skipping any unnecessary steps. |
| As a math teacher explaining the problem to a beginner, break the solution down into clear, digestible parts, using simple language and logical structure. |
| As a mathematical competitor in an Olympiad, look for elegant or clever shortcuts, tricks, or symmetry to reach the solution faster than using brute-force methods. |
| As a mathematician preparing to present your solution in a seminar, ensure every logical step is watertight and anticipate questions from an expert audience. |
| As a researcher writing a textbook, focus on clarity, pedagogical progression, and general principles that can apply beyond the immediate problem. |
| Use deductive reasoning to solve the mathematics problem by starting from known premises and building a logically rigorous sequence of steps that leads to the correct conclusion. |
| Apply first-principles thinking: break the problem down to its most basic definitions and axioms, and construct your solution without relying on pre-established shortcuts or formulas. |
| Break the problem into smaller sub-problems, solve them independently, and then integrate the partial results into a complete solution. |
| Apply a known theorem or identity carefully, making sure its conditions are satisfied in the current context before proceeding with the solution. |

Table 23: Variants of prompt applied to the Alpaca dataset

| |
|---|
| You are an assistant that should think through the problem step by step before answering. |
| As an assistant, your goal is to provide clear, logical reasoning behind every answer. |
| You're helping someone understand a complex issue—break it down into manageable parts. |
| Take your time. You are expected to consider various angles before responding. |
| You are a careful assistant. Avoid assumptions and stick to what the information supports. |
| Your job is to respond clearly and directly. Focus only on the key points. |
| Be thorough. As an assistant, you must evaluate all options before offering a suggestion. |
| You're an assistant trained to explain not just what, but why. |
| You are expected to analyze carefully before reaching a conclusion. |
| Keep your explanation structured. The person you're helping needs to follow your logic easily. |

Table 24: Variants of prompt applied to the DS-MVTec dataset

| |
|---|
| You are an industrial quality control inspector analyzing product images for potential defects. Review the image and determine if it meets manufacturing standards. |
| As a factory inspection expert, evaluate the product in the image for any signs of surface irregularities or damage that may affect usability. |
| You are a defect detection specialist examining product photographs. Assess the image and report whether a defect is present, providing a justification. |
| You are performing a visual inspection for a manufacturing audit. Examine the image and decide if the product displays any nonconformities. |
| Act as a certified industrial inspector evaluating high-resolution images of manufactured items. Judge if there are any quality issues present. |
| As an image-based inspection analyst, assess the product photo for any discrepancies compared to the expected standard. |
| Take on the role of a factory floor quality inspector reviewing image submissions. Determine if the item shown contains a production flaw. |
| You are a visual defect detection professional. Scrutinize the product image to evaluate for any scratches, cracks, or misalignments. |
| As a product evaluation specialist working from visual data, assess if the item in the image exhibits any indication of malfunction or imperfection. |
| Serve as an expert in machine vision inspection for industry. Determine whether the item in the provided image meets quality criteria. |

