# OpenReview forum: "Reinforcement Learning for Generalized Label Aggregation"
_ICLR.cc/2026/Conference — Submitted to ICLR 2026_

### Official Review · Reviewer_EUzf · 2025-10-31

**Soundness:** 2
**Presentation:** 3
**Contribution:** 2
**Rating:** 4
**Confidence:** 4

**Summary:**

This paper proposes a novel RL-based aggregation of LLM-generated data annotations. The authors constructed a dataset containing annotations and explanations using Qwen2.5-7B. Then, the authors used a recently developed RL framework, GRPO, to obtain an effective aggregator. The main focus of the authors was the development of the RL reward functions.

**Strengths:**

1. The research problem is practically interesting. I agree with the authors that the LLM-generated annotations are different from human-generated ones, and therefore, it is necessary to develop more effective aggregators.
2. The RL model of the aggregation problem is interesting and promising.
3. The authors constructed a new dataset for the research. The dataset contains well-known and widely-used previous datasets.

**Weaknesses:**

1. The problem setting is to aggregate annotations from different LLMs or one LLM with different personalization (Sec. 3.1). However, the dataset AGG was constructed using Qwen2.5-7B only. Although the authors mentioned a “highly scalable strategy” which was “significantly more practical”, I don’t see how that is true.

2. The reward functions are not well-explained. For example, I don’t see how the format reward and accuracy reward are calculated.

3. The authors decided to use GRPO as the reinforcement learning framework without clear reasons. Although they have mentioned stability, sample efficiency, etc., the argument is quite high-level and unconvincing. The details of the GRPO framework (Eq. 4) are not explained either.

4. The concept of “difficulty of an instance” is not intuitive. I don’t understand why the number of available annotations reflects difficulty. For example, an instance may have a large number of highly consistent annotations. I don’t think this case is difficult.

5. In the experiments, the authors used only the Qwen2.5-7B model. Considering the fact that the dataset AGG is generated using exactly the same model, I’m curious what would happen if the annotator LLMs are diverse and the aggregator LLM is different.

**Questions:**

See detailed comments (especially the ones in weaknesses).

---

> ### Author Response · Authors · 2025-12-04
>
> > W1: The dataset uses only Qwen2.5-7B.
>
> We appreciate the comment. While annotations are generated using Qwen2.5-7B, it is a representative mid-scale LLM widely used in practice. More importantly, as shown in Section 4.2 (Appendix D in previous version), RFAGG achieves consistent improvements when annotations come from diverse, non-Qwen LLMs. This demonstrates strong cross-model generalization and proves that RFAGG is not overfitting to the specific artifacts of the Qwen model.
>
>
> > W2: The reward functions are not well-explained
>
> We have revised Section 3.3 to clearly define both rewards:
> ​
> - $R_\text{format}$ is a rule-based binary signal that checks whether the output properly separates reasoning and final answer via a lightweight parser;
> - $R_\text{acc}$ is computed by extracting the final answer using predefined rules and comparing it to the gold label via exact match.
>
> Both are rule-based and deterministic and require no auxiliary models.
>
> > W3: The choice of GRPO lacks detailed justification
>
> We chose GRPO (Group Relative Policy Optimization) because it naturally handles multi-component reward signals like ours without requiring a value network (Critic). This significantly reduces memory usage and training instability, which is crucial when the "state" involves long contexts of multiple LLM annotations.
>
> > W4: Use annotation count as the proxy for difficulty, instead of answer consistency.
>
> We use annotation count as a measure of input complexity because it directly affects token length and computational load. In contrast, answer diversity does not reliably reflect difficulty: easy tasks may have high agreement but low discriminability, while hard tasks often show more diversity that can help distinguish correct answers. Moreover, high agreement may stem from bias and still be incorrect. Since this relationship is unstable and influenced by various factors, we base the curriculum on annotation count as a practical and scalable complexity indicator.
>
> > W5: The evaluation uses the same model for annotation and aggregation, raising questions about cross-model applicability.
>
> Our main evaluation in Section 4.2 shows that RFAGG surpasses other aggregation methods even when annotations come from diverse models while the aggregator remains fixed. This demonstrates its ability to effectively fuse heterogeneous annotation sources. Additionally, our new experiment in Appendix E, Table 3 evaluates RFAGG using different LLMs as aggregators. Notably, larger models do not always perform better, confirming that aggregation effectiveness depends on more than model size and highlighting the value of our approach.
>
>
> Appendix E. Table 3: Comparison with Different LLMs for Aggregation.
>
> | Dataset       | AGNews | DBPedia14 | Emotion | IMDB  | MultiNLI | SST2  | TREC  | WikiToxic | MMLU  | BoolQ | WiC  |
> |---------------|--------|-----------|---------|-------|----------|-------|-------|-----------|-------|-------|------|
> | ANNOTATION    | 80.0   | 94.8      | 58.0    | 95.0  | 82.8     | 93.1  | 81.4  | 78.8      | 72.0  | 82.2  | 66.4 |
> | Qwen2.5-7B    | 81.3   | 96.7      | 57.7    | 95.4  | 85.5     | 94.1  | 85.4  | 80.9      | 72.0  | 83.3  | 68.6 |
> | Qwen2.5-14B   | 77.4   | 87.6      | 55.6    | 93.5  | 84.1     | 90.9  | 82.4  | 81.4      | 78.0  | 82.7  | 69.1 |
> | MISTRAL-7B    | 68.1   | 78.4      | 46.8    | 76.6  | 60.6     | 73.9  | 64.8  | 59.3      | 32.0  | 66.2  | 44.9 |
> | GEMMA-3-12B   | 80.8   | 96.7      | 58.3    | 95.5  | 86.2     | 93.4  | 85.8  | 80.2      | 76.0  | 84.6  | 68.1 |
> | RFAGG         | **89.2** | **98.3**  | **65.0**| **96.0**| **86.8** | **96.1**| **91.8**| **85.7**  | **78.0**| **86.8**| **70.6**|

---

### Official Review · Reviewer_3jDf · 2025-11-01

**Soundness:** 2
**Presentation:** 2
**Contribution:** 2
**Rating:** 4
**Confidence:** 3

**Summary:**

This paper proposes RFAGG, a RL framework for aggregating annotations generated by LLMs. Unlike traditional aggregation methods that assume independent human annotators, RFAGG explicitly models both the labels and their accompanying justifications produced by LLMs.

Experiments on AGG and several out-of-domain datasets show that RFAGG substantially outperforms traditional aggregation baselines and untuned LLM aggregators. The framework achieves gains of up to 5.2% on AlpacaEval 2.

**Strengths:**

1. Reformulates label aggregation as a reinforcement learning problem, going beyond classical probabilistic or consensus-based formulations.

2. The proposed AGG dataset systematically covers multiple modalities and annotation types, providing a valuable benchmark.

3. The appendix includes prompts, data splits, and ablations that make replication feasible.

**Weaknesses:**

1. Outdated baselines, mostly pre-LLM (≤ 2019).
The experimental comparisons in Table 1 include only traditional label aggregation methods such as Majority Voting, Dawid–Skene (1979), CATD (2014), and several Bayesian variants from 2019 (BWA, IBCC, EBCC). These baselines represent the pre-LLM era of aggregation research. Since 2020, many new approaches have emerged that leverage LLM capabilities — e.g., self-consistency, debate-style or multi-agent aggregation, explanation-aware adjudication, and LLM-as-judge verification frameworks — which have substantially improved benchmark performance. The paper does not discuss or compare against any of these post-LLM aggregation methods, leaving its empirical positioning incomplete.

2. Table 1 compares only against a basic LLM (Qwen2.5-7B).
The only LLM baseline considered is the untuned Qwen2.5-7B-Instruct, which serves both as annotator and aggregator. No stronger or specialized LLMs are included. Consequently, it is unclear whether the reported gains hold for modern high-performing LLMs or under diverse architectures.

3. Limited discussion of recent LLM-alignment and reward-learning baselines.
Given that RFAGG uses reinforcement learning (GRPO) for aggregation, comparisons with newer alignment-oriented techniques — such as Direct Preference Optimization (DPO), Reinforcement Learning from AI Feedback (RLAF), or explanation-aware SFT — would strengthen the methodological justification. Without these, the necessity of using RL over simpler preference-based objectives remains partly unsubstantiated.

**Questions:**

The paper evaluates on widely used benchmarks such as AGNews, SST2, and MMLU. While these are well-established, their difficulty and ceiling performance have shifted dramatically with modern LLMs. For instance, models on MMLU-Pro already reach around 93%, whereas RFAGG reports only 78% on MMLU (Table 1).

Could the authors clarify why they chose to evaluate only on the original MMLU instead of more challenging or recent variants such as MMLU-Pro, MMLU-STEM, or DeepBench?

How should we interpret the 78% figure in a landscape where even base LLMs far exceed that on newer leaderboards?

---

> ### Author Response · Authors · 2025-12-04
>
> We thank the reviewer for raising important concerns. Below, we address each point with additional experimental evidence and clarifications.
>
> > W1: Post-LLM Methods
>
> To directly respond to this concern, we conducted an additional experiment comparing RFAGG against representative post-LLM aggregation methods in Appendix E, Table 4. As shown in the results, while these methods improve over naive annotation in most cases, RFAGG consistently outperforms all of them across datasets. This demonstrates that our approach provides a stronger learning signal than implicit or heuristic-based LLM aggregation techniques.
>
> Appendix E. Table 4: Comparison with Post-LLM Strategies.
>
> | Dataset           | AGNews | DBPedia14 | Emotion | IMDB  | MultiNLI | SST2  | TREC  | WikiToxic | MMLU  | BoolQ | WiC  |
> |-------------------|--------|-----------|---------|-------|----------|-------|-------|-----------|-------|-------|------|
> | SELF CONSISTENCY | 80.7   | 96.5      | 59.4    | 95.5  | 85.1     | 93.8  | 84.8  | 79.8      | 75.0  | 83.3  | 69.2 |
> | DEBATE STYLE      | 82.8   | 97.3      | 55.8    | 95.3  | 78.2     | 94.3  | 84.0  | 87.6      | 84.5  | 67.0  | 62.6 |
> | LLM-AS-JUDGE      | 81.0   | 96.1      | 55.5    | 95.2  | 80.5     | 94.4  | 81.4  | 82.7      | 74.0  | 84.0  | 60.0 |
> | RFAGG             | **89.2** | **98.3**  | **65.0**| **96.0**| **86.8** | **96.1**| **91.8**| **85.7**  | **78.0**| **86.8**| **70.6**|
>
>
> That said, we also note that Majority Voting (MV) remains a fundamental benchmark for aggregation capability. In fact, self-consistency can be viewed as a form of MV over generated answers, and its effectiveness is still largely evaluated relative to MV. Our goal is to improve the aggregation process itself, not replace it with multi-agent reasoning or verification. Thus, comparing against MV and Bayesian models (e.g., DS, EBCC) remains valid and meaningful for evaluating core aggregation performance.
>
> > W2: LLM Baseline Strength
>
> In Section 4.6, we show that when different LLMs are used as annotators, RFAGG still significantly improves over other aggregation baselines, demonstrating its robustness across annotation model types.
>
> Additionally, we have added a new experiment in Appendix E, Table 3, comparing RFAGG against multiple LLMs serving as aggregators. As reported in Table 3, larger models do not consistently yield better aggregation performance; for example, Qwen2.5-14B underperforms its 7B variant on several datasets. This highlights that model scale alone does not guarantee better consensus, and validates our claim that effective aggregation ability has room for improvement beyond mere model capacity.
>
> Appendix E. Table 3: Comparison with Different LLMs for Aggregation.
>
> | Dataset       | AGNews | DBPedia14 | Emotion | IMDB  | MultiNLI | SST2  | TREC  | WikiToxic | MMLU  | BoolQ | WiC  |
> |---------------|--------|-----------|---------|-------|----------|-------|-------|-----------|-------|-------|------|
> | ANNOTATION    | 80.0   | 94.8      | 58.0    | 95.0  | 82.8     | 93.1  | 81.4  | 78.8      | 72.0  | 82.2  | 66.4 |
> | Qwen2.5-7B    | 81.3   | 96.7      | 57.7    | 95.4  | 85.5     | 94.1  | 85.4  | 80.9      | 72.0  | 83.3  | 68.6 |
> | Qwen2.5-14B   | 77.4   | 87.6      | 55.6    | 93.5  | 84.1     | 90.9  | 82.4  | 81.4      | 78.0  | 82.7  | 69.1 |
> | MISTRAL-7B    | 68.1   | 78.4      | 46.8    | 76.6  | 60.6     | 73.9  | 64.8  | 59.3      | 32.0  | 66.2  | 44.9 |
> | GEMMA-3-12B   | 80.8   | 96.7      | 58.3    | 95.5  | 86.2     | 93.4  | 85.8  | 80.2      | 76.0  | 84.6  | 68.1 |
> | RFAGG         | **89.2** | **98.3**  | **65.0**| **96.0**| **86.8** | **96.1**| **91.8**| **85.7**  | **78.0**| **86.8**| **70.6**|
>
> > W3: Comparison with Modern Alignment & Reward Learning Methods
>
> We acknowledge that DPO, RLAF, and SFT are effective for alignment, but they focus on preference modeling rather than aggregation. Our goal is to maximize final accuracy by fusing multiple noisy annotations, a distinct task where direct comparison with those methods is not applicable. Additionally, we already include an SFT baseline in Figure 3, and as shown, the results are significantly worse than RFAGG.
>
>
>
> > Q: Performance on MMLU and Dataset Choice
>
> We selected datasets from tinyBenchmarks[1], a curated subset designed to evaluate LLMs with fewer examples and controlled difficulty.
> The 78% accuracy reported in Table 1 reflects performance using Qwen2.5-7B, not state-of-the-art LLMs. As noted in the Qwen2 Technical Report[2], Qwen2.5-7B achieves 70.3% on MMLU. Larger models (e.g., 72B) reach 84.2%, but our goal is to evaluate aggregation improvement over base model performance, not absolute ceiling.
>
> [1] Polo F M, Weber L, Choshen L, et al. tinyBenchmarks: evaluating LLMs with fewer examples[J]. arXiv preprint arXiv:2402.14992, 2024.
>
> [2] Team Q. Qwen2 technical report[J]. arXiv preprint arXiv:2407.10671, 2024, 2(3).

---

### Official Review · Reviewer_ZfEw · 2025-11-01

**Soundness:** 3
**Presentation:** 3
**Contribution:** 3
**Rating:** 6
**Confidence:** 2

**Summary:**

The paper introduces an RL framework (RFAGG) for LLM annotations aggregator. The aggregator conditions on the input plus K annotations (label + justification) and generates a reasoned aggregation. Rewards combine format correctness, answer accuracy, and an entropy-weighted bonus for high-disagreement items; optimization uses GRPO with a curriculum over annotation count. The authors construct an AGG dataset (mostly text classification/multiple-choice; some reasoning and a vision-inspection suite) using Qwen-2.5 variants to simulate diverse annotators. RFAGG outperforms majority vote and probabilistic crowd models on in-domain tests and shows generalization to other domains.

**Strengths:**

+ Clear problem framing: aggregating correlated LLM “votes” with usable rationales, not just labels.
+ Reward design is intuitive and targeted at the true goal (accuracy + conflict resolution), not just imitation.
+ Simple, plausible training recipe (GRPO + curriculum over #annotations).
+ Broad empirical sweep with consistent gains over MV/DS-style baselines; nice to see open-ended (AlpacaEval 2) and a vision domain.
+ Ablations support the components (RL > SFT, reward shaping, curriculum).

**Weaknesses:**

- Reward details under-specified. The entropy threshold τ, reward scaling, and sensitivity analyses are not clarified. It would be good to see how performance varies with τ and with different ways to compute annotation entropy when labels are open-ended.
- The LLM-aggregator baseline is just “untuned Qwen-2.5-7B,” which might be too weak. Using a stronger (e.g. a larger proprietary model) / reasoning-optimized model or a stronger straw-man that explicitly mirrors the three reward factors (format compliance, answer accuracy, and disagreement/entropy handling) would add confidence.
- Compute / cost / stability. No training-time or sampling-cost details (e.g., group size for GRPO, #rollouts) are given here; reproducibility depends on those knobs.

**Questions:**

Apart from my concerns in the weaknesses:
- How is τ chosen in the entropy bonus, and how sensitive are results to τ and reward scaling? Are there any per-task tuning?
- In the curriculum, what are typical kmin/kcurr and how many steps per stage? Does performance keep improving as K (number of annotations) grows, or does it saturate?
- Does RFAGG assume binary labels? How does it handle multi-class or ordinal labels? Are richer, non-binary inputs (e.g., annotator probabilities/confidence) expected to improve performance?

---

> ### Author Response · Authors · 2025-12-04
>
> We thank the reviewer for the thoughtful feedback. Below, we provide detailed responses to your points.
>
> > W1 & Q1: Reward Design and Sensitivity
>
> We do not introduce per-task tuning; all training and inference steps use a unified set of parameters (mostly default). We set $\tau = 0.2$ and found it effective across tasks. We utilized this strategy rather than extensive parameter tuning to demonstrate that: 1) Our method generalizes well without hypersensitive tuning, and 2) We do not have enough computing resources to conduct parameter tuning.
>
> While careful tuning per task might yield marginally better results, the current performance already validates the contribution of our work.
>
> Regarding open-ended tasks: These are not used during training, as they do not admit rule-based accuracy rewards or reliable exact-match evaluation. Therefore, $\tau$ only applies to closed-set classification tasks where labels are discrete and entropy is well-defined. However, our experiments verify that the reasoning capabilities learned via this method generalize effectively to open-ended tasks at inference time.
>
>
>
> > W2: LLM-Aggregator Baseline:
>
> To directly evaluate whether model scale alone can improve aggregation performance, we conducted an additional experiment comparing multiple LLM aggregators. As shown in Appendix E, Table 3, larger models do not consistently outperform smaller ones. For example, Qwen2.5-14B underperforms its 7B counterpart on several datasets. This suggests that simply increasing model capacity is insufficient for achieving better aggregation. In contrast, RFAGG incorporates structured reasoning and reward-guided learning, consistently outperforming all LLM-based baselines.
>
>
> Appendix E. Table 3: Comparison with Different LLMs for Aggregation.
>
> | Dataset       | AGNews | DBPedia14 | Emotion | IMDB  | MultiNLI | SST2  | TREC  | WikiToxic | MMLU  | BoolQ | WiC  |
> |---------------|--------|-----------|---------|-------|----------|-------|-------|-----------|-------|-------|------|
> | ANNOTATION    | 80.0   | 94.8      | 58.0    | 95.0  | 82.8     | 93.1  | 81.4  | 78.8      | 72.0  | 82.2  | 66.4 |
> | Qwen2.5-7B    | 81.3   | 96.7      | 57.7    | 95.4  | 85.5     | 94.1  | 85.4  | 80.9      | 72.0  | 83.3  | 68.6 |
> | Qwen2.5-14B   | 77.4   | 87.6      | 55.6    | 93.5  | 84.1     | 90.9  | 82.4  | 81.4      | 78.0  | 82.7  | 69.1 |
> | MISTRAL-7B    | 68.1   | 78.4      | 46.8    | 76.6  | 60.6     | 73.9  | 64.8  | 59.3      | 32.0  | 66.2  | 44.9 |
> | GEMMA-3-12B   | 80.8   | 96.7      | 58.3    | 95.5  | 86.2     | 93.4  | 85.8  | 80.2      | 76.0  | 84.6  | 68.1 |
> | RFAGG         | **89.2** | **98.3**  | **65.0**| **96.0**| **86.8** | **96.1**| **91.8**| **85.7**  | **78.0**| **86.8**| **70.6**|
>
> Additionally, we compared RFAGG with other advanced post-LLM baselines (see Table 4 below), further verifying the effectiveness of our approach.
>
> Appendix E. Table 4: Comparison with Post-LLM Strategies.
>
> | Dataset           | AGNews | DBPedia14 | Emotion | IMDB  | MultiNLI | SST2  | TREC  | WikiToxic | MMLU  | BoolQ | WiC  |
> |-------------------|--------|-----------|---------|-------|----------|-------|-------|-----------|-------|-------|------|
> | SELF CONSISTENCY | 80.7   | 96.5      | 59.4    | 95.5  | 85.1     | 93.8  | 84.8  | 79.8      | 75.0  | 83.3  | 69.2 |
> | DEBATE STYLE      | 82.8   | 97.3      | 55.8    | 95.3  | 78.2     | 94.3  | 84.0  | 87.6      | 84.5  | 67.0  | 62.6 |
> | LLM-AS-JUDGE      | 81.0   | 96.1      | 55.5    | 95.2  | 80.5     | 94.4  | 81.4  | 82.7      | 74.0  | 84.0  | 60.0 |
> | RFAGG             | **89.2** | **98.3**  | **65.0**| **96.0**| **86.8** | **96.1**| **91.8**| **85.7**  | **78.0**| **86.8**| **70.6**|
>
>
> > W3: Computational and Training Details
>
> Training required approximately two days on a server with eight H800 GPUs, using a group size of 8. Other hyperparameters followed the default settings of the ms-swift library.
>
> > Q2: In the curriculum, what are typical kmin/kcurr and how many steps per stage? Does performance keep improving as K (number of annotations) grows, or does it saturate?
>
> We start with $K_{min}=3$ and increment to $K_{max}=10$. It is difficult to precisely measure saturation because the model is evolving simultaneously with the curriculum.
>
> > Q3: Does RFAGG assume binary labels? How does it handle multi-class or ordinal labels? Are richer, non-binary inputs (e.g., annotator probabilities/confidence) expected to improve performance?
>
> RFAGG treats aggregation as a sequence generation task. It is not limited to binary labels; it generates the target token directly. For ordinal or multi-class tasks, the model learns to output the specific class name or index justified by the prompt context.

---

### Official Review · Reviewer_udxn · 2025-11-05

**Soundness:** 2
**Presentation:** 2
**Contribution:** 3
**Rating:** 4
**Confidence:** 3

**Summary:**

The paper proposes an LLM based method for label aggregation, i.e., creating a label for an input which has been labeled/annotated by multiple LLMs. The approach is based on training an LLM using RL to jointly model the label as well as the justification, based on those provided by the annotator LLMs. For this, the paper proposes a reward function which has components addressing the aggregation format, accuracy and complexity. The paper combines an existing RL objective (GRPO), with a curriculum training on those inputs with less number of annotations (and hence simpler) to be aggregated. The paper also introduces an AGG datasets whose training dataset is constructed from eight well known benchmark datasets and the test data combines 5 other datasets covering a broader spectrum of tasks. The annotator pool was simulated in the experiments using different personalities of prompts to one of two specific models. The paper claims significant improvements of their method over previous non-LLM label aggregation methods, and also using general purpose LLMs for aggregation.

I wish to clarify that I have previously reviewed this paper, and while there are some improvements from the previous version, the paper has not been significantly changed.

**Strengths:**

The paper tackles the important problem of label aggregation using general purpose LLMs which are increasingly being used for specialized computational and modeling tasks. The RL and curriculum based method is potentially useful.

**Weaknesses:**

1. The exposition of the proposed method in the paper is somewhat lacking in detail and clarity: the various rewards (e.g. format reward and accuracy reward)  are not defined formally, and it is not clear how they are obtained (is an LLM used?). While Appendix  C has a description of the MDP, there is no reference to this appendix from the main paper.
2. The overall approach uses an existing RL training (GRPO) with some fairly natural rewards and a curriculum. No analytical analysis or justification is provided.
3. Further, from the ablations in Fig. 3, it seems that removing either the curriculum or complex rewards does not diminish the performance significantly.

**Questions:**

1. Missing references from the main paper to relevant appendices.
2. Use parenthesized citations in Section 2, and on line 55.
3. The results in Table 3 seem implausible - all seven baselines have exactly the same performance on ARC.

---

> ### Author Response · Authors · 2025-12-04
>
> Thank you for your careful review and valuable feedback on our paper. We have made significant revisions compared to the previous version, and we believe most of your concerns have been addressed in the updated manuscript. Below, we provide detailed responses to your specific points.
>
> > Q1: Missing references from the main paper to relevant appendices
>
> Thank you for pointing this out. We have now added explicit cross-references within the main text to all relevant supplementary materials to ensure better navigability.
>
> > Q2: Use parenthesized citations in Section 2, and on line 55
>
> We appreciate you catching this inconsistency. We have revised the citations to strictly follow the parenthesized style throughout the manuscript.
>
> > Q3: The results in Table 3 seem implausible
>
> We appreciate this keen observation. To clarify, we have re-verified all entries against our raw evaluation logs and confirm their correctness. The identical results occurred because, on the specific ARC dataset subset we used, the annotator LLMs exhibited extremely high correlation (they either all answered correctly or all answered incorrectly). Consequently, all consensus-based baselines (MV, DS, etc.) collapsed to the exact same prediction. RFAGG was able to break this ceiling by leveraging the justifications to identify cases where the majority was hallucinating.
>
> To fully address this concern, we have uploaded a supplementary file containing detailed results for baseline methods on this dataset, including per-instance predictions and aggregation results.
>
>
> > W1: Details of  format reward and accuracy reward and appendix of MDP
>
> We have revised Section 3.3 to include explicit mathematical definitions for both the Format Reward $R_\text{format}$ and the Accuracy Reward $R_\text{acc}$. As stated in the updated manuscript:
> - $R_\text{format}$ is a rule-based function computed via a simple parser that checks if the output contains properly separated reasoning and final answer sections.
> - $R_\text{acc}$ is also rule-based, computed by extracting the final answer using predefined rules and comparing it to the gold label via exact match.
>
> These are deterministic, lightweight functions; no additional LLM is used to compute these rewards, ensuring efficiency and reproducibility.
> We have now added a sentence at the end of Section 3.3 explicitly stating: "The MDP formulation can be found in Appendix C." This directly links the reader to the formal framework.
>
> > W2: Justification for the overall approach
>
> Our work targets a critical problem in LLM research: training a model to aggregate information from diverse, noisy sources. Our novel application of GRPO to jointly model labels and justifications introduces a multi-faceted reward function tailored to aggregation. It ensures structured outputs, accuracy, and intelligent conflict resolution in high-disagreement cases, which differs fundamentally from standard RL approaches for dialogue or instruction following. The curriculum strategy starts with simpler annotations to mimic natural learning progression, preventing the model from being overwhelmed by complex contexts early in training.
>
> > W3: Ablation study
>
> While the impact of removing specific components (like the curriculum or specific rewards) varies across datasets, the full RFAGG method often yields the best results, particularly on difficult datasets like MMLU and WiC. These results verify that the proposed training method provides superior performance and better generalization ability compared to the ablated versions.

---

### Author Response · Authors · 2025-12-04

We extend our sincere gratitude to all reviewers for their careful evaluation and constructive feedback throughout the review and discussion phases. We have provided detailed responses to address each of the reviewers' concerns. All newly added content in the revised manuscript is highlighted in red, and all modified content is marked in blue for clarity.

This work demonstrates that it is possible to build a generalized model using a robust reinforcement training strategy. Notably, we found that the model generalizes effectively to diverse domains, including visual and open-ended tasks. We believe this work makes a timely and impactful contribution by establishing a generalized aggregation model. This capability will become increasingly vital with the rapid advancement of LLMs, particularly within multi-agent research.

Given the novelty, thorough evaluation, and broad applicability of our approach, we hope the committee will find RFAGG a strong candidate for acceptance at ICLR.

The Authors

---

### Meta-Review · Area_Chair_Cpbk · 2026-01-05

**Summary:**

This paper presents a method for RLing an LLM to perform label aggregation: look at annotations from "workers" and produce an aggregate judgment. A model is trained via RL to aggregate labels to the correct label, with an objective combining format reward and upweighting correctness on more complex (high annotator entropy) examples.

Strengths:

- Interesting "LLM-forward" approach for the label aggregation task

- Comparison with a number of existing label aggregation methods

- Validation across a number of datasets in both in-domain and out-of-domain settings

Weaknesses:

- Relatively small gains in the generalization sets. The in-domain sets are not apples-to-apples with baselines as the training method is quite different. I am quite unconvinced by the response to reviewer ZfEw, as this is comparing zero-shot aggregation with a method trained on each dataset specifically, a very unfair comparison.

- Lack of strong LLM baselines: zero-shot with weak LLMs, or approaches added in rebuttal like debate which aren't explained at all and which are hard to validate as strong baselines.

**Reviewer Concerns:**

Clarity issues / lack of explanation of rewards: This appears to be addressed in the revision.

Weak LLMs: The rebuttal addresses this somewhat, but the other LLMs used are largely weaker than Qwen (except for the 14B model).  There's not much discussion of the issue that Qwen is both the annotator and the aggregator.

Weak baselines (debate, other LLM aggregators): These approaches added in rebuttal, like debate, aren't explained at all and are hard to validate as strong baselines.

Unclear gains of method given ablations: I'm not super convinced by this. The performance is pretty shaky.

Choice of GRPO: In my view, this does not need to be justified, so I think this is resolved. I don't think this paper is claiming that GRPO is necessary vs. PPO or other RL methods.

**Reviewer Scores:**

I think in general, the issues around clarity have been resolved. However, clarity issues have a way of then leading to other objections to the paper, as points that were initially unclear become targets for objection. I can't confidently say that reviewers would increase scores on the basis of this.

Personally I think the additional experiments are unconvincing. These are all on the in-domain setting. If I'm understanding correctly, RFAGG is trained on the AGG test set, but none of the other methods are. These make it a not apples-to-apples comparison.

Furthermore, the baselines added during rebuttal are lacking in detail. I can't tell how well these baselines were really implemented, or if they were implemented well.  I think in order for the paper to deliver a really convincing win, it needs to convince the reader that the baselines are strong and give a clearer view of why this method works and where it works.

Specifc criticisms:

udxn: underspecified implementation, ablations don't change much.

ZfEw: underspecified implementation, outdated/weak baselines

3jDf: outdated/weak baselines. (I disregard weakness 3.)

EUzf: underspecified implementation, weak baselines

I don't see the rebuttals being very effective on these points in general. So I don't think the overall tenor of the scores and analysis would change.

---

### Decision · Program_Chairs · 2026-01-26

Reject